# Weighted $L^1$ and $L^0$ Regularization Using Proximal Operator Splitting Methods

**Zewude A. Berkessa**  *zewude.berkessa@oulu.fi*
*Research Unit of Mathematical Sciences*
*University of Oulu*

**Patrik Waldmann**  *patrik.waldmann@oulu.fi*
*Research Unit of Mathematical Sciences*
*University of Oulu*

**Reviewed on OpenReview:** *https://openreview.net/forum?id=9m2k96cDMK*

## Abstract

This paper develops a joint weighted $L^1$- and $L^0$-norm (WL1L0) regularization method by leveraging proximal operators and translation mapping techniques to mitigate the bias introduced by the $L^1$-norm in applications to high-dimensional data. A weighting parameter $\alpha$ is incorporated to control the influence of both regularizers. Our broadly applicable model is nonconvex and nonsmooth, but we show convergence for the alternating direction method of multipliers (ADMM) and the strictly contractive Peaceman–Rachford splitting method (SCPRSM). Moreover, we evaluate the effectiveness of our model on both simulated and real high-dimensional genomic datasets by comparing with adaptive versions of the least absolute shrinkage and selection operator (LASSO), elastic net (EN), smoothly clipped absolute deviation (SCAD) and minimax concave penalty (MCP). The results show that WL1L0 outperforms the LASSO, EN, SCAD and MCP by consistently achieving the lowest mean squared error (MSE) across all datasets, indicating its superior ability to handling large high-dimensional data. Furthermore, the WL1L0-SCPRSM also achieves the sparsest solution. Julia code for the WL1L0-ADMM and WL1L0-SCPRSM is available at `https://github.com/ZewAB/WL1L0-ADMM-and-SCPRSM`.

## 1 Introduction

High-dimensional statistics is a rapidly growing field of research that focuses on statistical analysis in the presence of a large number of variables or predictors ($p$), often much larger than the sample size ($n$). For example, high-throughput measurements in genomics contain thousands or millions of variables, such as single nucleotide polymorphism (SNP) markers and gene expression data for each individual. In such settings, traditional statistical methods often fail due to issues like overfitting, multicollinearity and computational complexity. In recent years, a number of regularization methods have been developed that impose a penalty on the size of the regression coefficients, which encourages sparsity and reduces the number of variables in the model (Fan et al., 2011; Fan & Lv, 2010; Heinze et al., 2018). Sparse learning techniques are essential in analyzing high-dimensional data for increased prediction accuracy, reduced computational complexity and enhanced interpretability of the results (Bühlmann & Van De Geer, 2011; Giraud, 2015; Wainwright, 2019).

Among various sparsity-inducing methods, the $L^1$ regularizer (also known as the least absolute shrinkage and selection operator (LASSO)) stands out for its convex nature and computational efficiency (Tibshirani, 1996). It adds a penalty term to the loss function proportional to the absolute value of the regression coefficients ($L^1$-norm), which tends to shrink the coefficients towards zero and force some coefficients to exactly zero. Shrinking these coefficients helps to avoid overfitting, which can happen when a model memorizes the training data too well and does not perform well on new data. The LASSO improves the accuracy of predictions on

unseen data by simultaneously selecting features and mitigating overfitting. However, when coefficients are being shrunk, bias is introduced due to the bias-variance trade-off that is unavoidable in statistical learning. Furthermore, the LASSO tends to favor keeping larger coefficients over smaller ones which can lead to a bias towards larger coefficients in the model estimation process (Hastie et al., 2015).

In the specific context of genomic data, where the goal is often to identify genes associated with certain traits or diseases, this inaccurate selection can lead to the inclusion of incorrect genes in the model. This also poses a risk in terms of impaired prediction, as the estimated coefficients of the selected genes contribute to predicting the trait of interest (Fan et al., 2014b; Fan & Li, 2001; Johnstone & Titterington, 2009; Toloşi & Lengauer, 2011). Hence, the LASSO requires the fulfillment of the irrepresentable condition to obtain valid estimations (Zhao & Yu, 2006). In cases where the underlying datasets fail to meet this condition, the LASSO method may not accurately select the appropriate variables, leading to incorrect discoveries and wrong conclusions. In practice, implementing the irrepresentable condition can be challenging. Studies show that nonconvex regularizers such as SCAD and MCP reduce bias and have better prediction properties than the $L^1$ regularizer (Bertsimas et al., 2020; Fan & Li, 2001; Zhang, 2010).

On the other hand, $L^0$ regularization, which is also known as best subset selection (Hocking & Leslie, 1967), directly penalizes the number of non-zero coefficients in the model. It encourages sparsity, meaning it tends to produce models with fewer non-zero coefficients without any shrinkage. This results in a model that only includes the most relevant variables, simplifying the model and potentially improving its predictive performance by reducing overfitting. However, finding the optimal subset of variables using the $L^0$-norm is computationally expensive because the $L^0$-norm is nonconvex. While $L^1$ regularization is commonly used because of its convex nature, the $L^0$-norm is computationally expensive and often intractable, and hence not frequently used on large data sets (Hastie et al., 2020).

Another regularization method is $L^2$, also known as ridge regularization, which shrinks the coefficients towards zero without eliminating any of them completely (Hoerl & Kennard, 1970). Unlike the LASSO, ridge regression produces dense estimated regression coefficients, which means it does not perform feature selection. It reduces the size of the coefficients but does not drive any of them to exactly zero. Furthermore, the elastic net (EN) is another regularization technique that combines the properties of ridge regression and LASSO regression (Zou & Hastie, 2005). It is particularly useful for datasets with many features, especially when some are highly correlated. The LASSO may struggle with grouped variable selection, often picking just one from correlated variables, whereas EN improves feature selection in such cases (Hastie et al., 2015).

In this paper, we propose combining $L^1$ and $L^0$ regularization into a method denoted WL1L0 for improved prediction and variable selection. $L^1$ regularization encourages sparsity by shrinking some coefficients to zero, which helps reduce overfitting and is computationally efficient due to its convex properties. On the other hand, $L^0$ regularization enforces strict sparsity, directly penalizing the number of non-zero coefficients (i.e., eliminating variables with negligible impact), leading to more interpretable models. This synergy offers a better balance between interpretability and predictive accuracy, particularly in high-dimensional settings like genomics. Furthermore, since $L^0$ is an unbiased estimator and $L^1$ often introduces biases in estimation, $L^0$ can be regarded as debiasing $L^1$ in this setting. We achieve this goal by using a common regularization parameter and introducing a weight parameter that balances the importance of the two regularization methods.

We address the computational challenges that arise from optimizing the $L^0$-norm by using proximal splitting methods, translation mapping and the efficient optimization algorithms ADMM and SCPRSM. The prediction and sparsity properties of the WL1L0 method is evaluated on one simulated and two real genomic datasets and compared with the popular LASSO, EN, SCAD and MCP regularizers.

## 2 Related Work

In the rapidly evolving landscape of technology and data, prediction has become a cornerstone for making informed decisions across various domains. Regularization techniques are pivotal in enhancing the performance and generalizability of predictive models, particularly when dealing with complex datasets and high-dimensional data. By imposing penalties on the model parameters, regularization helps prevent over-

fitting, ensuring that the model captures the underlying patterns in the data. In this section, we will review key related works on regularization methods, highlighting significant advancements and methodologies.

We start by introducing a standard regression model

$$\boldsymbol{y} = \boldsymbol{X}\boldsymbol{b} + \boldsymbol{\epsilon}, \tag{1}$$

where $\boldsymbol{y} \in \mathbb{R}^n$ is the response vector, $\boldsymbol{X} \in \mathbb{R}^{n \times p}$ is the predictor matrix, $\boldsymbol{b} \in \mathbb{R}^p$ is the vector of regression coefficients, and $\boldsymbol{\epsilon} \in \mathbb{R}^n$ is a noise (error) vector. For a vector $\boldsymbol{b}$, we write the $q$-norm notation as

$$||\boldsymbol{b}||_q = \begin{cases} \sum_i \mathbf{1}\left(b_i \neq 0\right), & \text{if } q = 0, \\ \left(\sum_i |b_i|^q\right)^{1/q}, & \text{if } 0 < q < \infty, \\ \max_i |b_i|, & \text{if } q = \infty, \end{cases}$$

where $i = 1, \cdots, p$. Here, the $||\boldsymbol{b}||_0$ is the $L^0$-norm that is the number of nonzero elements in $\boldsymbol{b}$. It is noteworthy that the $L^0$-norm does not meet the criteria of a norm, specifically lacking the homogeneity property (Beck, 2017). Despite this, the term is widely used in the literature, and for the sake of consistency, we will retain its adoption.

Ridge regression, also known as Tikhonov regularization was introduced by Hoerl & Kennard (1970), uses an $L^2$ penalty term that shrinks all the coefficients and reduces their magnitudes. The ridge regression can be formulated as

$$\hat{\boldsymbol{b}} = \operatorname*{argmin}_{\boldsymbol{b}} ||\boldsymbol{y} - \boldsymbol{X}\boldsymbol{b}||_2^2 + \lambda ||\boldsymbol{b}||_2^2, \tag{2}$$

where $\lambda > 0$ is regulazation parameter need to be tuned. This method is particularly effective in addressing multicollinearity in linear regression models. However, it does not necessarily set any coefficients to zero. Hence, ridge regression does not produce a sparse solution of estimated coefficients. On the other hand, LASSO regression

$$\hat{\boldsymbol{b}} = \operatorname*{argmin}_{\boldsymbol{b}} ||\boldsymbol{y} - \boldsymbol{X}\boldsymbol{b}||_2^2 + \lambda ||\boldsymbol{b}||_1, \tag{3}$$

incorporates an $L^1$ regularization penalty, which encourages sparsity in the solution by setting some coefficients exactly to zero (Tibshirani, 1996).

Other penalty functions are introduced to provide a balance between inducing sparsity and reducing estimation bias, aiming to solve the optimization problem as

$$\hat{\boldsymbol{b}} = \operatorname*{argmin}_{\boldsymbol{b}} ||\boldsymbol{y} - \boldsymbol{X}\boldsymbol{b}||_2^2 + P_\lambda(\boldsymbol{b}). \tag{4}$$

For example, the smoothly clipped absolute deviation (SCAD) penalty function was introduced by Fan & Li (2001) as an improvement over LASSO regularization, particularly for bias reduction. The SCAD penalty function is defined as

$$P_\lambda^{SCAD}(\boldsymbol{b}) = \begin{cases} \lambda |\boldsymbol{b}| & \text{if } |\boldsymbol{b}| \leq \lambda, \\ \frac{-|\boldsymbol{b}|^2 + 2a\lambda|\boldsymbol{b}| - \lambda^2}{2(a-1)} & \text{if } \lambda < |\boldsymbol{b}| \leq a\lambda, \\ \frac{(a+1)\lambda^2}{2} & \text{if } |\boldsymbol{b}| > a\lambda, \end{cases} \tag{5}$$

where $\lambda > 0$ and $a > 0$ are unknown parameters. Fan & Li (2001) suggested that $a = 3.7$ is a good choice for various problems, and $\lambda$ needs to be tuned.

The minimax concave penalty (MCP) is another type of penalty function introduced by Zhang (2010). The MCP penalty function is defined as

$$P_\lambda^{MCP}(\boldsymbol{b}) = \begin{cases} \lambda |\boldsymbol{b}| - \frac{\boldsymbol{b}^2}{2a} & \text{if } |\boldsymbol{b}| \leq \lambda a, \\ \frac{a\lambda^2}{2} & \text{if } |\boldsymbol{b}| > a\lambda. \end{cases} \tag{6}$$

According to the estimation theorems of Zhang (2010), $a = 3$ is a good choice for MCP, and $\lambda$ still needs to be tuned. MCP was developed to address the estimation bias of the LASSO and is generally easier to optimize computationally compared to SCAD.

Both SCAD and MCP aim to eliminate unimportant variables while preserving important ones, achieving the 'oracle property' as the sample size grows ($n \to \infty$). They both asymptotically select the correct model and produce normal, accurate coefficient estimates. MCP is effective with many sparse predictor groups but struggles with tightly clustered non-zero coefficients while SCAD has weaker grouping behavior compared to MCP (Ogutu & Piepho, 2014). We maintain the use of $a = 3.7$ for SCAD and $a = 3$ for MCP throughout the paper.

For the $L^0$ regularization (best subset selection (BSS)), $P_\lambda(\boldsymbol{b})$ can be written as

$$P_\lambda^{BSS}(\boldsymbol{b}) = \lambda \sum_i^p \mathbf{1}\left(b_i \neq 0\right). \tag{7}$$

Exact optimization of problem (4) with the $L^0$-norm, as defined in (7), is challenging because incorporating (7) into the objective function results in a non-differentiable and non-convex problem. For example, Louizos et al. (2017) propose a method for optimizing a relaxed version of the $L^0$ norm for parametric models using a distribution called the hard concrete distribution (Maddison et al., 2016), which facilitates gradient-based optimization.

Yun et al. (2019) use a family of $M$-estimators with trimmed regularization for general high-dimensional problems. The trimmed regularization problem can be formulated for LASSO as

$$\begin{aligned}
\hat{\boldsymbol{b}}, \hat{\boldsymbol{\pi}} = \operatorname*{argmin}_{\boldsymbol{b}, \boldsymbol{\pi}} &\; ||\boldsymbol{y} - \boldsymbol{X}\boldsymbol{b}||_2^2 + \lambda \sum_{i=1}^p \pi_i |b_i| \\
\text{subject to} \quad &\mathbf{1}^\top \boldsymbol{\pi} \geq p - h, \\
&\boldsymbol{\pi} \in [0, 1],
\end{aligned} \tag{8}$$

where $h$ denotes the trimming parameter, which must be appropriately tuned, for instance, through cross-validation.

Another example is the elastic net (EN) regression which combines both $L^1$ and $L^2$ regularization penalties, providing a balanced approach to prediction accuracy on future data and model interpretation in linear regression models. It is formulated as

$$\hat{\boldsymbol{b}} = \operatorname*{argmin}_{\boldsymbol{b}} ||\boldsymbol{y} - \boldsymbol{X}\boldsymbol{b}||_2^2 + \lambda_1 ||\boldsymbol{b}||_1 + \lambda_2 ||\boldsymbol{b}||_2^2, \tag{9}$$

which has two regularization parameters $\lambda_1$ and $\lambda_2$ to tune (Zou & Hastie, 2005). The LAVA regression model is based on the splitting of the regression component into one sparse and one dense part $\boldsymbol{b} = \boldsymbol{c} + \boldsymbol{d}$ and thereby obtaining the following optimization problem

$$\hat{\boldsymbol{c}}, \hat{\boldsymbol{d}} = \operatorname*{argmin}_{\boldsymbol{c}, \boldsymbol{d}} ||\boldsymbol{y} - \boldsymbol{X}(\boldsymbol{c} + \boldsymbol{d})||_2^2 + \lambda_1 ||\boldsymbol{c}||_1 + \lambda_2 ||\boldsymbol{d}||_2^2, \tag{10}$$

where the resulting estimator $\hat{\boldsymbol{b}} = \hat{\boldsymbol{c}} + \hat{\boldsymbol{d}}$ (Chernozhukov et al., 2017). The key difference between EN and LAVA is that EN performs variable selection (i.e., is dominated by the $L^1$-norm), whereas LAVA is always dense (i.e., is dominated by the $L^2$-norm). Waldmann (2021) developed a proximal operator algorithm based on the LAVA regularization method that jointly performs $L^1$- and $L^2$-norm regularization.

Ziyin & Wang (2023) propose a method called *spred*, for optimizing generic differentiable objectives with an $L^1$ constraint using a reparametrization. The method is proposed to effectively bridge the gap between sparsity in deep learning and conventional statistical learning by providing a principled way to optimize $L^1$ constraints in complex nonlinear settings. For example, one can apply the *spred* parametrization to the sparse component of $\boldsymbol{b}_s$ given the LASSO loss $||\boldsymbol{y} - \boldsymbol{X}\boldsymbol{b}_s||_2^2 + 2\kappa ||\boldsymbol{b}_s||_1$. The equivalent *spred* loss is then

$$\hat{\boldsymbol{c}}, \hat{\boldsymbol{d}} = \operatorname*{argmin}_{\boldsymbol{c}, \boldsymbol{d}} ||\boldsymbol{y} - \boldsymbol{X}(\boldsymbol{c} \odot \boldsymbol{d})||_2^2 + \kappa(||\boldsymbol{d}||_2^2 + ||\boldsymbol{c}||_2^2), \tag{11}$$

where $\hat{\boldsymbol{b}} = \hat{\boldsymbol{c}} \odot \hat{\boldsymbol{d}}$, $\odot$ denotes the element-wise product, and $\kappa$ is the $L^1$ regularization strength parameter that needs to be tuned to achieve the best sparsity-performance trade-off.

## 3   Theoretical Background

Large parts of the theory behind our approach follows from (Bertsekas, 2016) and (Beck, 2017). For an extended real-valued function $f : \mathbb{R}^p \to [-\infty, \infty]$, we define the following:

(a) The domain of $f$ is the set
$$\text{dom}(f) = \{\boldsymbol{b} \in \mathbb{R}^p : f(\boldsymbol{b}) < \infty\}.$$

(b) $f$ is proper if $\text{dom}(f) \neq \varnothing$ and $f$ is never $-\infty$.

(c) The epigraph of $f$ is defined by
$$\text{epi}(f) = \{(\boldsymbol{b}, a) \in \mathbb{R}^p \times \mathbb{R} : f(\boldsymbol{b}) \leq a\}.$$

(d) The function $f$ is closed if its epigraph is closed.

(e) $f$ is called lower semicontinuous at $\boldsymbol{b} \in \mathbb{R}^p$ if
$$f(\boldsymbol{b}) \leq \liminf_{k \to \infty} f(\boldsymbol{b}^{(k)})$$
for any sequence $\{\boldsymbol{b}^{(k)}\}_{k \geq 1} \subseteq \mathbb{R}^p$ for which $\boldsymbol{b}^{(k)} \to \boldsymbol{b}$ as $k \to \infty$.

(f) For any $\eta \in \mathbb{R}$, the $\eta$-level set of a function $f$ is the set
$$\text{Lev}(f, \eta) = \{\boldsymbol{b} \in \mathbb{R}^p : f(\boldsymbol{b}) \leq \eta\}.$$

(g) A proper function $f$ is called coercive if
$$\lim_{||\boldsymbol{b}|| \to \infty} f(\boldsymbol{b}) = \infty.$$

For any set $\mathbb{S} \subseteq \mathbb{R}^p$ and any point $\boldsymbol{b} \in \mathbb{R}^p$, the distance from $\boldsymbol{b}$ to $\mathbb{S}$ is defined as $D(\boldsymbol{b}, \mathbb{S}) := \inf\{||\boldsymbol{m} - \boldsymbol{b}||, \boldsymbol{m} \in \mathbb{S}\}$, and $D(\boldsymbol{b}, \mathbb{S}) = \infty$ for all $\boldsymbol{b}$ when $\mathbb{S} = \varnothing$.

A proper closed and coercive function $f$ attains its minimal value over $\mathbb{S}$ for a nonempty closed set satisfying $\mathbb{S} \cap \text{dom}(f) = \emptyset$. Moreover, a closed coercive function possesses a minimizer on any closed set that has a nonempty intersection with the domain of the function (Beck, 2017). For an extended real-valued function $f : \mathbb{R}^p \to [-\infty, \infty]$, the following three claims are equivalent:

i $f$ is lower semicontinuous.

ii $f$ is closed.

iii For any $\eta \in \mathbb{R}$, the level set
$$\text{Lev}(f, \eta) = \{\boldsymbol{b} \in \mathbb{R}^p : f(\boldsymbol{b}) \leq \eta\}$$
is closed.

The proof of these claims can be found in (Beck, 2017), see Theorem 2.6.

### 3.1   Subdifferentials of Nonconvex and Nonsmooth Functions

Subdifferentials are important in analyzing complex functions, especially when dealing with nonsmooth and nonconvex functions. Following Clarke et al. (2008) and Mordukhovich (2006), we explore subdifferentiability.

Let $g : \mathbb{R}^p \to (-\infty, +\infty]$ be a proper and lower semicontinuous function. Then

(i) For a given $\boldsymbol{b} \in \text{dom } g$, the Fréchet subdifferential of $g$ at $\boldsymbol{b}$, denoted by $\hat{\partial}g(\boldsymbol{b})$, is the set of all vectors $\boldsymbol{u} \in \mathbb{R}^p$ which satisfy

$$\liminf_{\boldsymbol{m} \to \boldsymbol{b}} \frac{g(\boldsymbol{m}) - g(\boldsymbol{b}) - \langle \boldsymbol{u}, \boldsymbol{m} - \boldsymbol{b} \rangle}{||\boldsymbol{m} - \boldsymbol{b}||} \geq 0,$$

and we set $\hat{\partial}g = \varnothing$ when $\boldsymbol{b} \notin \text{dom } g$.

(ii) The limiting-subdifferential, or simply the subdifferential, of $g$ at $\boldsymbol{b}$, written by $g(\boldsymbol{b})$, is defined by

$$\partial g(\boldsymbol{b}) := \{\boldsymbol{u} \in \mathbb{R}^p : \exists \boldsymbol{b}^{(k)} \to \boldsymbol{b}, g(\boldsymbol{b}^{(k)}) \to g(\boldsymbol{b}) \text{ and } \boldsymbol{u}^{(k)} \in \hat{\partial}g(\boldsymbol{b}^{(k)}) \overset{k \to \infty}{\longrightarrow} \boldsymbol{u}\},$$

where $\hat{\partial}g(\boldsymbol{b}) \subset \partial g(\boldsymbol{b})$ for all $\boldsymbol{b} \in \mathbb{R}^p$.

(iii) A point $\boldsymbol{b}^*$ is called critical point or stationary point of $g$ if it satisfies $0 \in \partial g(\boldsymbol{b}^*)$.

Please refer to Wu et al. (2021) for generalized subdifferentials of the $L^0$, with its regular subdifferentials provided in Le (2013).

## 3.2 The Kurdyka–Łojasiewicz Inequality and its Property

The Kurdyka-Łojasiewicz (KŁ) inequality deals with the behavior of certain functions near their critical points. It is an important tool for analyzing the convergence of nonconvex nonsmooth optimization problems (Attouch et al., 2010; 2013; Bolte et al., 2014). We now revice the KŁ property.

Let $g : \mathbb{R}^p \to (-\infty, +\infty]$ be a proper lower semicontinuous function. Then,

(a) The function $g : \mathbb{R}^p \to \mathbb{R} \cup \{+\infty\}$ is said to have the KŁ property at $\boldsymbol{b}^* \in \text{dom } \partial g$ if there exist $\eta \in (0, +\infty]$, a neighborhood $U$ of $\boldsymbol{b}^*$, and a continuous concave function $\phi : [0, \eta) \to \mathbb{R}^+$ such that

   (i) $\phi(0) = 0$,
   (ii) $\phi$ is continuously differentiable on $(0, \eta)$,
   (iii) $\forall a \in (0, +\infty]$, $\phi'(a) > 0$,
   (IV) For all $\boldsymbol{b} \in U \cap \{g(\boldsymbol{b}^*) < g(\boldsymbol{b}) < g(\boldsymbol{b}^*) + \eta\}$, the KŁ property holds:

   $$\phi'(g(\boldsymbol{b}) - g(\boldsymbol{b}^*))d(0, \partial g(\boldsymbol{b})) \geq 1.$$

(b) Proper lower semicontinuous functions which satisfy the KŁ inequality at each point of dom $\partial g$ are called KŁ functions. Examples of KŁ functions include $||\boldsymbol{b}||_1$, $||\boldsymbol{b}||_0$, and $||\boldsymbol{y} - \boldsymbol{X}\boldsymbol{b}||_2^2$. For more examples, please refer to (Attouch et al., 2010; 2013; Bolte et al., 2014; Yashtini, 2022).

## 3.3 Proximal Operators

Proximal operators are a fundamental concept in optimization, especially for problems involving non-smooth or non-convex functions, which are increasingly common in a wide range of real-world applications (Fukushima & Mine, 1981; Kaplan & Tichatschke, 1998; Parikh & Boyd, 2013). A proximal operator, denoted as $\text{prox}_f(\boldsymbol{u})$, aims to find a point closer to $\boldsymbol{u}$ that also minimizes a specific objective function, $f(\boldsymbol{v})$ in a specific optimization subproblem. This subproblem is assumed to be more manageable to solve than the original problem. The proximal operator can be mathematically expressed as

$$\text{prox}_f(\boldsymbol{u}) = \underset{\boldsymbol{v}}{\text{argmin}} \{f(\boldsymbol{v}) + (1/2)|||\boldsymbol{v} - \boldsymbol{u}||_2^2\}, \tag{12}$$

where $\boldsymbol{u}$ and $\boldsymbol{v}$ are vectors of length $p$. Here, $\text{prox}_f(\boldsymbol{u})$ is a point that compromises between minimizing $f$ and being close to $\boldsymbol{u}$. Note that the right-hand side of (12) is strongly convex, hence there is a unique minimizer for every $\boldsymbol{u} \in \mathbb{R}^p$. Introducing the parameter $\gamma > 0$ that represents a trade-off parameter between the two terms $\boldsymbol{v}$ and $\boldsymbol{u}$ yields a scaled version of (12), in which $\frac{1}{2}$ is replaced by $\frac{1}{2\gamma}$. The proximal operator has useful properties (Beck, 2017), one of which is its behavior when applied to affine functions, as shown below.

**Lemma 1** *For any affine function $f(\boldsymbol{u}) = \langle \boldsymbol{m}, \boldsymbol{u} \rangle + a$, where $\boldsymbol{m} \in \mathbb{R}^p$ is a fixed vector and $a \in \mathbb{R}$, then for any $\boldsymbol{u} \in \mathbb{R}^p$, the proximal operator defined in (12) reduces to a simple translation of the vector $\boldsymbol{u}$ by $\boldsymbol{m}$. Specifically,*

$$prox_f(\boldsymbol{u}) = \boldsymbol{u} - \boldsymbol{m}, \tag{13}$$

*which represents a translation mapping.*

The proof of Lemma 1 is provided in Appendix B.1. In accordance with Lemma 1, one defines a translation function as a function that incorporates a standard additive term which is expressed as $\mathcal{T}_{\boldsymbol{m}}(\boldsymbol{u}) = f(\boldsymbol{u}+\boldsymbol{m}) - \boldsymbol{m}$. Another important property arises in the context of separable sum functions $f(\boldsymbol{u}, \boldsymbol{m}) = g(\boldsymbol{u}) + h(\boldsymbol{m})$, where the proximal operator is written as $\mathrm{prox}_f(\boldsymbol{u}, \boldsymbol{m}) = \mathrm{prox}_g(\boldsymbol{u}) + \mathrm{prox}_h(\boldsymbol{m})$. For proximal operators in the framework of $L^0$, please refer to (Attouch et al., 2013; Bolte et al., 2014; Beck, 2017).

## 4    Methodological Framework

To achieve higher sparsity than the EN, one can use the least squares loss function with $\ell_1$ and $\ell_0$ norm constraints which can be formulated as

$$\hat{\boldsymbol{b}} = \operatorname*{argmin}_{\boldsymbol{b}} ||\boldsymbol{y} - \boldsymbol{X}\boldsymbol{b}||_2^2 \quad \text{s.t.} \quad ||\boldsymbol{b}||_1 \leq t, \quad ||\boldsymbol{b}||_0 \leq s, \tag{14}$$

where $\hat{\boldsymbol{b}}$ represents the estimate of the vector of regression coefficients, $t$ is a constant threshold and $s$ is the desired level of sparsity (i.e., the maximum number of nonzero coefficients). The optimization problem (14) can be expressed in the Lagrangian form as

$$\hat{\boldsymbol{b}} = \operatorname*{argmin}_{\boldsymbol{b}} ||\boldsymbol{y} - \boldsymbol{X}\boldsymbol{b}||_2^2 + \lambda_1 ||\boldsymbol{b}||_1 + \lambda_2 ||\boldsymbol{b}||_0, \tag{15}$$

where $\lambda_1 > 0$ and $\lambda_2 > 0$ are the regularization parameters.

We formulate our proposed method by building upon the flexible, penalty-based framework introduced in (15) that uses the parameters $\lambda_1$ and $\lambda_2$ to control both the size of the coefficients and the sparsity in a more flexible and nuanced manner, allowing for a broader range of model behaviors compared to the constant threshold and the strict subset selection enforced by formulation (14). Here, the exact relationship between $t$ and $\lambda_1$ and between $s$ and $\lambda_2$ is data-dependent.

We now extend (15) by introducing a weight parameter $\alpha \in (0, 1)$, a common regularization parameter $\lambda > 0$ and reformulate the problem as

$$\hat{\boldsymbol{b}} = \operatorname*{argmin}_{\boldsymbol{b}} ||\boldsymbol{y} - \boldsymbol{X}\boldsymbol{b}||_2^2 + \lambda \left( \alpha ||\boldsymbol{b}||_1 + (1-\alpha)||\boldsymbol{b}||_0 \right). \tag{16}$$

Furthermore, we formulate the problem (16) following the LAVA method. Hence, we split the regression component $\boldsymbol{b}$ into the sparse components $\boldsymbol{c}$ and $\boldsymbol{d}$ and separately assign penalties $L^1$ and $L^0$ to them and therefore obtain

$$\text{WL1L0: } \hat{\boldsymbol{c}}, \hat{\boldsymbol{d}} = \operatorname*{argmin}_{\boldsymbol{c}, \boldsymbol{d}} ||\boldsymbol{y} - \boldsymbol{X}(\boldsymbol{c} + \boldsymbol{d})||_2^2 + \lambda \left( \alpha ||\boldsymbol{c}||_1 + (1-\alpha)||\boldsymbol{d}||_0 \right). \tag{17}$$

It is here useful to point out that EN combines $L^1$-norm and $L^2$-norm regularization which leads to variable selection while being less sensitive to correlated predictors than the LASSO. In contrast, LAVA is dominated by the $L^2$-norm, leading to dense models without feature selection. WL1L0 combines $L^1$ and $L^0$ regularization, providing stricter feature selection by explicitly controlling the number of non-zero coefficients, resulting in more precise sparsity than the EN. See Appendix A for additional insights into the problem and discussion.

## 5    Optimization Algorithms

Alternating direction method of multipliers (ADMM) is often used as a benchmark algorithm for splitting problems due to its efficiency and flexibility (Boyd et al., 2011). One of its notable strengths is its ability to

handle large-scale optimization problems by decomposing them into smaller, more manageable subproblems. This decomposition not only simplifies the problem-solving process but also allows for parallel processing of these subproblems. The strictly contractive Peaceman–Rachford splitting method (SCPRSM) is a variant of the classical Peaceman–Rachford splitting method (PRSM) (He et al., 2014). Similar to ADMM, PRSM is an operator splitting technique used to solve optimization problems. SCPRSM is a further extension that ensures convergence by imposing a strict contraction condition. Although ADMM is more widely used in practice, SCPRSM is a more specialized method that guarantees convergence through strict contraction. While SCPRSM emphasizes strict contraction, ADMM may exhibit slower convergence under certain conditions. Therefore, we implement our proposed method using both ADMM and SCPRSM frameworks based on the augmented Lagrangian method that combines the original objective function with the constraints of the optimization problem into a single function. Here, the augmented Lagrangian's advantage lies in enabling the study of convergence for the proposed methods without requiring assumptions like strict convexity (Boyd et al., 2011).

The introduction of two variables ($\boldsymbol{c}$ and $\boldsymbol{d}$) instead of one ($\boldsymbol{b}$) in (17) increases the dimensionality of the optimization problem, adding complexity to the theoretical analysis. Therefore, we first study the convergence properties of the problem in (16). We then reformulate problem (16) to establish its convergence in the ADMM and SCPRSM frameworks. The optimization model for (16) can be formulated as

$$\hat{\boldsymbol{b}} = \operatorname*{argmin}_{\boldsymbol{b}} \{f(\boldsymbol{b}) + g(\boldsymbol{b})\} \quad \Longleftrightarrow \quad \hat{\boldsymbol{b}} = \operatorname*{argmin}_{\boldsymbol{b}} \{f(\boldsymbol{b}) + g(\boldsymbol{u})\}$$
$$\text{subject to} \quad \boldsymbol{b} = \boldsymbol{u},$$
(18)

where $f(\boldsymbol{b}) = \|\boldsymbol{y} - \boldsymbol{X}(\boldsymbol{b})\|_2^2$ is the loss function and $g(\boldsymbol{b}) = \lambda\left(\alpha\|\boldsymbol{b}\|_1 + (1-\alpha)\|\boldsymbol{b}\|_0\right)$ is the penalty function. We now write the augmented Lagrangian function corresponding to (18) as

$$L_\gamma(\boldsymbol{b}, \boldsymbol{u}, \boldsymbol{z}) = f(\boldsymbol{b}) + g(\boldsymbol{u}) + \boldsymbol{z}^T(\boldsymbol{b} - \boldsymbol{u}) + \frac{\gamma}{2}\|\boldsymbol{b} - \boldsymbol{u}\|_2^2,$$
(19)

where $\boldsymbol{z}$ is a dual variable or Lagrange multiplier and $\gamma > 0$ is a learning rate. Here, $\boldsymbol{b}$ and $\boldsymbol{u}$ are the primal variables.

## 5.1 Method of Multipliers and ADMM Framework

The method of multipliers jointly minimizes the two primal variables whereas the ADMM efficiently solves optimization problems by alternately updating primal and dual variables, effectively decomposing complex problems into manageable subproblems (Boyd et al., 2011). A more convenient scaled form of (19) can be obtained by completing the square with the dual variable $\boldsymbol{z}$ and the residual $\boldsymbol{b} - \boldsymbol{u}$ in the augmented Lagrangian. This allows the term $\boldsymbol{z}^T(\boldsymbol{b} - \boldsymbol{u}) + \frac{\gamma}{2}|\boldsymbol{b} - \boldsymbol{u}|_2^2$ to be rewritten as $\frac{\gamma}{2}\|\boldsymbol{b} - \boldsymbol{u} + \frac{1}{\gamma}\boldsymbol{z}\|_2^2 - \frac{\gamma}{2}\|\boldsymbol{z}\|_2^2$. Introducing the scaled dual variable $\boldsymbol{m} = \frac{1}{\gamma}\boldsymbol{z}$, the scaled form of the augmented Lagrangian becomes

$$L_\gamma(\boldsymbol{b}, \boldsymbol{u}, \boldsymbol{m}) = f(\boldsymbol{b}) + g(\boldsymbol{u}) + \frac{\gamma}{2}\|\boldsymbol{b} - \boldsymbol{u} + \boldsymbol{m}\|_2^2 - \frac{\gamma}{2}\|\boldsymbol{m}\|_2^2.$$
(20)

This scaled form is better suited for implementing ADMM and SCPRSM schemes with proximal operators (Parikh & Boyd, 2013). The method of multipliers for (20) can be written as

$$(\boldsymbol{b}^{(k+1)}, \boldsymbol{u}^{(k+1)}) := \operatorname*{argmin}_{\boldsymbol{b}, \boldsymbol{u}} L_\gamma(\boldsymbol{b}^{(k)}, \boldsymbol{u}^{(k)}, \boldsymbol{m}^{(k)}),$$
(21)

$$\boldsymbol{m}^{(k+1)} := \boldsymbol{m}^{(k)} + \boldsymbol{b}^{(k+1)} - \boldsymbol{u}^{(k+1)}.$$
(22)

The method of multipliers is generally not an implementable method since the primal update step (21) can be as hard to solve as the original problem (Beck, 2017; Boyd et al., 2011). To overcome this challenge, ADMM employs an iterative approach in the primal update step. In this approach, $\boldsymbol{b}$ and $\boldsymbol{u}$ are updated sequentially in an alternating fashion, which is why the method is called the alternating direction method of multipliers.

An iterative scheme for the ADMM associated with (20) becomes

$$\boldsymbol{b}^{(k+1)} := \operatorname*{argmin}_{\boldsymbol{b}} L_\gamma(\boldsymbol{b}^{(k)}, \boldsymbol{u}^{(k)}, \boldsymbol{m}^{(k)}), \tag{23a}$$

$$\boldsymbol{u}^{(k+1)} := \operatorname*{argmin}_{\boldsymbol{u}} L_\gamma(\boldsymbol{b}^{(k+1)}, \boldsymbol{u}^{(k)}, \boldsymbol{m}^{(k)}), \tag{23b}$$

$$\boldsymbol{m}^{(k+1)} := \boldsymbol{m}^{(k)} + \boldsymbol{b}^{(k+1)} - \boldsymbol{u}^{(k+1)}. \tag{23c}$$

## 5.2 SCPRSM Framework

The difference between ADMM and PRSM in terms of convergence can be explained through the contraction properties of their iterative sequences. The iterative sequence generated by ADMM is strictly contractive with respect to a given solution set, whereas the sequence generated by PRSM is contractive, but not strictly contractive (He et al., 2014; Corman & Yuan, 2014; He et al., 2002). To address the lack of strict contraction in PRSM, He et al. (2014) proposed incorporating a relaxation factor $r > 0$ into the Lagrange multiplier update steps, thus developing a strictly contractive Peaceman-Rachford splitting method (SCPRSM). This modification ensures that the iterative sequence becomes strictly contractive, improving convergence properties. The studies show that SCPRSM outperforms that ADMM generally leads to faster convergence compared to ADMM (Li & Yuan, 2015; Li et al., 2021). The iterative scheme of the SCPRSM associated with the augmented Lagrangian function (20) is written as

$$\boldsymbol{b}^{(k+1)} := \operatorname*{argmin}_{\boldsymbol{b}} L_\gamma(\boldsymbol{b}^{(k)}, \boldsymbol{u}^{(k)}, \boldsymbol{m}^{(k)}), \tag{24a}$$

$$\boldsymbol{m}^{(k+\frac{1}{2})} := \boldsymbol{m}^{(k)} + r(\boldsymbol{b}^{(k+1)} - \boldsymbol{u}^{(k)}), \tag{24b}$$

$$\boldsymbol{u}^{(k+1)} := \operatorname*{argmin}_{\boldsymbol{u}} L_\gamma(\boldsymbol{b}^{(k+1)}, \boldsymbol{u}^{(k)}, \boldsymbol{m}^{(k+\frac{1}{2})}), \tag{24c}$$

$$\boldsymbol{m}^{(k+1)} := \boldsymbol{m}^{(k+\frac{1}{2})} + r(\boldsymbol{b}^{(k+1)} - \boldsymbol{u}^{(k+1)}), \tag{24d}$$

where the parameter $r \in (0,1)$ is a relaxation factor. In addition to a relaxation factor $r$, an important distinction between SCPRSM and ADMM is the presence of an intermediate update for the multipliers, represented as $\boldsymbol{m}^{(k+\frac{1}{2})}$ in the SCPRSM scheme. This step ensures a balanced handling of the vectors $\boldsymbol{b}$ and $\boldsymbol{u}$, thereby leading to a contractive iteration sequence that guarantees convergence to the solution of the original optimization problem (He et al., 2014; Li & Yuan, 2015; Peaceman & Rachford, 1955). The iterative scheme of ADMM (23a - 23c) can be further simplified as

$$
\begin{aligned}
\boldsymbol{b}^{(k+1)} &:= \operatorname*{argmin}_{\boldsymbol{b}} \{f(\boldsymbol{b}^{(k)}) + \frac{\gamma}{2}||\boldsymbol{b}^{(k)} - (\boldsymbol{u}^{(k)} - \boldsymbol{m}^{(k)})||_2^2\}, \\
\boldsymbol{u}^{(k+1)} &:= \operatorname*{argmin}_{\boldsymbol{u}} \{g(\boldsymbol{u}^{(k)}) + \frac{\gamma}{2}||\boldsymbol{u}^{(k)} - (\boldsymbol{b}^{(k+1)} + \boldsymbol{m}^{(k)})||_2^2\}, \\
\boldsymbol{m}^{(k+1)} &:= \boldsymbol{m}^{(k)} + \boldsymbol{b}^{(k+1)} - \boldsymbol{u}^{(k+1)}.
\end{aligned}
\tag{25}
$$

With proximal operators, we can now rewrite (25) as

$$
\begin{aligned}
\boldsymbol{b}^{(k+1)} &:= \operatorname{prox}_{f\gamma}(\boldsymbol{u}^{(k)} - \boldsymbol{m}^{(k)}), \\
\boldsymbol{u}^{(k+1)} &:= \operatorname{prox}_{g\gamma}(\boldsymbol{b}^{(k+1)} + \boldsymbol{m}^{(k)}), \\
\boldsymbol{m}^{(k+1)} &:= \boldsymbol{m}^{(k)} + \boldsymbol{b}^{(k+1)} - \boldsymbol{u}^{(k+1)}.
\end{aligned}
\tag{26}
$$

Similarly, the proximal version of Equations (24a - 24d) can be written as

$$
\begin{aligned}
\boldsymbol{b}^{(k+1)} &:= \operatorname{prox}_{f\gamma}(\boldsymbol{u}^{(k)} - \boldsymbol{m}^{(k)}), \\
\boldsymbol{m}^{(k+\frac{1}{2})} &:= \boldsymbol{m}^{(k)} + r(\boldsymbol{b}^{(k+1)} - \boldsymbol{u}^{(k)}), \\
\boldsymbol{u}^{(k+1)} &:= \operatorname{prox}_{g\gamma}(\boldsymbol{b}^{(k+1)} + \boldsymbol{m}^{(k+\frac{1}{2})}), \\
\boldsymbol{m}^{(k+1)} &:= \boldsymbol{m}^{(k+\frac{1}{2})} + r(\boldsymbol{b}^{(k+1)} - \boldsymbol{u}^{(k+1)}).
\end{aligned}
\tag{27}
$$

### 5.3 Convergence Analysis

Now, we delve into the convergence properties of the iterative schemes of ADMM (23) and SCPRSM (24). This analysis will elucidate the conditions under which these methods converge and the nature of the solutions they yield. Specifically, the convergence of the proposed method is established through the following theorems.

**Theorem 1** *Let the sequences $\{\boldsymbol{b}^{(k)}, \boldsymbol{u}^{(k)}, \boldsymbol{m}^{(k)}\}_{k=0}^{\infty}$ be generated by the ADMM scheme (23a - 23c) and its Lagrangian is given by (20). Then the following three conditions hold:*

*(a) **Sufficient decrease condition:** For each iteration step $k$, $\exists \delta_1 > 0$ such that*

$$L_\gamma(\boldsymbol{b}^{(k+1)}, \boldsymbol{u}^{(k+1)}, \boldsymbol{m}^{(k+1)}) - L_\gamma(\boldsymbol{b}^{(k)}, \boldsymbol{u}^{(k)}, \boldsymbol{m}^{(k)}) \leq -\delta_1 ||\boldsymbol{b}^{(k+1)} - \boldsymbol{b}^{(k)}||_2^2.$$

*(b) **Boundness condition:** The sequences $\{\boldsymbol{b}^{(k)}, \boldsymbol{u}^{(k)}, \boldsymbol{m}^{(k)}\}_{k=0}^{\infty}$ are bounded and its Lagrangian $L_\gamma(\boldsymbol{b}^{(k)}, \boldsymbol{u}^{(k)}, \boldsymbol{m}^{(k)})$ is lower bounded.*

*(c) **Convergence:** The Lagrangian in (20) is a Kurdyka-Łojasiewicz (KŁ) function, then the corresponding sequence $\{\boldsymbol{b}^{(k)}, \boldsymbol{u}^{(k)}, \boldsymbol{m}^{(k)}\}$ converges to a unique stationary point $\{\boldsymbol{b}^{(*)}, \boldsymbol{u}^{(*)}, \boldsymbol{m}^{(*)}\}$.*

Note that the function $f(\boldsymbol{b}) = ||\boldsymbol{y} - \boldsymbol{X}\boldsymbol{b}||_2^2$ is a continuously differentiable function with respect to $\boldsymbol{b}$. Its gradient is computed as $\nabla f(\boldsymbol{b}) = -2\boldsymbol{X}^T\boldsymbol{y} + 2\boldsymbol{X}^T\boldsymbol{X}\boldsymbol{b}$. Then the Lipschitz constant for the gradient of the function $f(\boldsymbol{b})$ can be computed as

$$\begin{aligned}
||\nabla f(\boldsymbol{b}^{(k)}) - \nabla f(\boldsymbol{b}^{(k+1)})|| &\leq 2||\boldsymbol{X}^T\boldsymbol{X}||\,||\boldsymbol{b}^{(k)} - \boldsymbol{b}^{(k+1)}|| \\
&\leq 2\lambda_{\max}(\boldsymbol{X}^T\boldsymbol{X})||\boldsymbol{b}^{(k)} - \boldsymbol{b}^{(k+1)}|| \\
&= l_f ||\boldsymbol{b}^{(k)} - \boldsymbol{b}^{(k+1)}||,
\end{aligned} \tag{28}$$

where $||\boldsymbol{X}^T\boldsymbol{X}||$ is the largest eigenvalue of $\boldsymbol{X}^T\boldsymbol{X}$ computed as $\lambda_{\max}(\boldsymbol{X}^T\boldsymbol{X})$. We denote the Lipschitz gradient constant as $l_f = 2\lambda_{\max}(\boldsymbol{X}^T\boldsymbol{X})$. The partial derivative of the Lagrangian (20) with respect to $\boldsymbol{b}$ is given by

$$\partial_{\boldsymbol{b}} L_\gamma(\boldsymbol{b}, \boldsymbol{u}, \boldsymbol{m}) = \nabla f(\boldsymbol{b}) + \gamma(\boldsymbol{b} - \boldsymbol{u} + \boldsymbol{m}), \tag{29}$$

and the second partial derivative is $\dfrac{\partial^2 L_\gamma(\boldsymbol{b}, \boldsymbol{u}, \boldsymbol{m})}{\partial \boldsymbol{b}^2} = 2\boldsymbol{X}^T\boldsymbol{X} + \gamma\boldsymbol{I}$ which is positive definite. This implies that the Lagrangian is strongly convex with respect to $\boldsymbol{b}$. We will frequently use the properties of the Lipschitz gradient constant of $f(\boldsymbol{b})$ and and the strong convexity of $L_\gamma(\boldsymbol{b}, \boldsymbol{u}, \boldsymbol{m})$ with respect to $\boldsymbol{b}$ in the proof (see Appendix B.2).

**Theorem 2** *Let the sequences $\{\boldsymbol{b}^{(k)}, \boldsymbol{u}^{(k)}, \boldsymbol{m}^{(k)}\}_{k=0}^{\infty}$ be generated by the scheme (24a - 24d) and its Lagrangian is given by (20). Then conditions (a)-(c) in Theorem 1 hold.*

The proof of Theorem 2 is found in Appendix B.3.

In conclusion, the sufficient decreasing and boundedness conditions are satisfied when the learning rate $\gamma > \max\{\dfrac{2l_f^2}{\rho}, l_f\}$ in both Theorems 1 and 2. In practice, choosing $\gamma$ involves a trade-off that requires careful consideration (see Section 5.5). The sufficient decreasing condition can be verified for the mean squared error (MSE) loss because the update step inherently minimizes the loss, ensuring it decreases as the number of iterations increases.

### 5.4 Implementation

The key idea in the WL1L0 optimization problem is to split the variable $\boldsymbol{b} = \boldsymbol{c} + \boldsymbol{d}$ into two components: $\boldsymbol{c}$, subject to $L^1$-norm regularization, and $\boldsymbol{d}$, subject to $L^0$-norm regularization. This decouples the optimization of $\boldsymbol{c}$ and $\boldsymbol{d}$ within the loss function $f(\boldsymbol{c} + \boldsymbol{d}) = ||y - X(\boldsymbol{c} + \boldsymbol{d})||^2$ allowing separate updates for each part.

The decoupling is achieved through defining two translation functions, which manage the updates efficiently. The translation functions are defined as $\mathcal{T}_v(u) = f(u + v) - v$ and $\mathcal{T}_u(v) = f(v + u) - u$. These enable alternating updates between $c$ and $d$, using the current estimates of the other variable. By leveraging these translations, the loss function $f(c + d)$ is effectively split, allowing the proximal operators to handle both $L^1$- and $L^0$-norm regularizations. Hence, for WL1L0-ADMM, the updates are made in six steps, alternating between the two primal variables $u$ and $v$, with corresponding dual variables $m$ and $w$. The steps are:

$$
\begin{aligned}
c^{(k+1)} &:= \text{prox}_{\mathcal{T}_v(u)\gamma}(u^{(k)} - m^{(k)}), \\
u^{(k+1)} &:= \text{prox}_{g\gamma}(c^{(k+1)} + m^{(k)}), \\
m^{(k+1)} &:= m^{(k)} + c^{(k+1)} - u^{(k+1)}, \\
d^{(k+1)} &:= \text{prox}_{\mathcal{T}_u(v)\delta}(v^{(k)} - w^{(k)}), \\
v^{(k+1)} &:= \text{prox}_{h\delta}(d^{(k+1)} + w^{(k)}), \\
w^{(k+1)} &:= w^{(k)} + d^{(k+1)} - v^{(k+1)}.
\end{aligned}
\tag{30}
$$

For WL1L0-SCPRSM, the updates are made in eight steps as

$$
\begin{aligned}
c^{(k+1)} &:= \text{prox}_{\mathcal{T}_v(u)\gamma}(u^{(k)} - m^{(k)}), \\
m^{(k+\frac{1}{2})} &:= m^{(k)} + r(c^{(k+1)} - u^{(k)}), \\
u^{(k+1)} &:= \text{prox}_{g\gamma}(c^{(k+1)} + m^{(k+\frac{1}{2})}), \\
m^{(k+1)} &:= m^{(k+\frac{1}{2})} + r(c^{(k+1)} - u^{(k+1)}), \\
d^{(k+1)} &:= \text{prox}_{\mathcal{T}_u(v)\delta}(v^{(k)} - w^{(k)}), \\
w^{(k+\frac{1}{2})} &:= w^{(k)} + r(d^{(k+1)} - v^{(k)}), \\
v^{(k+1)} &:= \text{prox}_{h\delta}(d^{(k+1)} + w^{(k+\frac{1}{2})}), \\
w^{(k+1)} &:= w^{(k+\frac{1}{2})} + r(d^{(k+1)} - v^{(k+1)}).
\end{aligned}
\tag{31}
$$

Here, $\text{prox}_{g\gamma}(c + m)$ is the proximal operator for $L^1$, which is the soft-thresholding function with learning rate $\gamma$ defined as

$$
\text{prox}_{g\gamma}(c + m) = \mathcal{S}_\gamma(c + m) = \max(0, |c + m| - \gamma)\text{sgn}(c + m),
\tag{32}
$$

and $\text{prox}_{h\delta}(d + w) = \mathcal{H}_{\sqrt{2\delta}}(d + w)$ is the proximal operator for $L^0$, which is hard thresholding operator defined as

$$
\mathcal{H}_{\sqrt{2\delta}}(d + w) = \begin{cases}
0, & \text{if } |d + w| < \sqrt{2\delta}, \\
d + w, & \text{if } |d + w| > \sqrt{2\delta}, \\
\{0, d + w\}, & \text{if } |d + w| = \sqrt{2\delta}.
\end{cases}
\tag{33}
$$

The iterations are terminated when convergence is reached according to $\|(c^{(k)} + d^{(k)}) - (u^{(k)} + v^{(k)})\|_\infty \leq \beta(1 + \|m^{(k)} + w^{(k)}\|_\infty)$ for tolerance parameter $\beta$ which was set to $10^{-5}$.

For comparison purposes, we also implement the LASSO, SCAD, MCP and EN methods using the proximal ADMM and SCPRSM schemes (see Appendix D). Note that the convergence analysis of (30) and (31) is straightforward from (23) and (24). However, the introduction of translation functions and the variables $c$ and $d$ increases the dimensionality of the optimization problem, making the theoretical analysis very extensive. Therefore, we omit the detailed proof of the algorithm that involves the translation functions.

## 5.5 Determining the Learning Rate

Choosing the learning rate (step size) is crucial for efficiency and proper convergence of optimization algorithms. There are two main methods for determining the learning rates $\gamma$ and $\delta$ (Beck, 2017; Bertsekas, 2016; Boyd & Vandenberghe, 2004): 1. Backtracking line-search: This method adjusts the learning rate iteratively based on specific criteria. However, it is both computationally expensive and time-consuming, as it requires

multiple evaluations of the objective function and its gradient during the search process. These repeated evaluations increase the overall computational load, particularly in high-dimensional problems. 2. Constant learning rate: In contrast, this method uses a fixed learning rate throughout the whole optimization process. It is computationally simpler and avoids the time overhead associated with frequent adjustments, making it more efficient in many scenarios.

We adopt the constant learning rate approach, using $\gamma^k = \frac{1}{||\boldsymbol{X}||_2}$ for all $k$, where $||\boldsymbol{X}||_2$ is the operator norm on the training set defined as

$$||\boldsymbol{X}||_2 := \max_{||\boldsymbol{b}||_2=1} ||\boldsymbol{X}\boldsymbol{b}||_2. \tag{34}$$

Equivalently, $||\boldsymbol{X}||_2$ is the maximum singular value of $\boldsymbol{X}$ ($\sigma_{\max}(\boldsymbol{X})$), which measures the maximum amount by which the matrix $\boldsymbol{X}$ can stretch a vector $\boldsymbol{b}$ relative to its original length (Horn & Johnson, 2013). The supremum-based definition can be applied in (34), particularly in infinite-dimensional contexts (Bhatia, 1997). However, in finite dimensions, the maximum and supremum coincide for the operator norm defined in (34). By setting the learning rate to $\frac{1}{||\boldsymbol{X}||_2}$, we ensure that the learning rate is scaled appropriately relative to the maximum possible stretch of $\boldsymbol{X}$. We use the same formula for $\delta$.

### 5.6 Bayesian Optimization for Hyperparameter Tuning

Tuning the regularization parameter $\lambda$, the weight parameter $\alpha$ and the relaxation factor $r$ via cross-validation or grid search can be computationally expensive. Bayesian Optimization (BO) is a more advanced, data-driven approach which offers a probabilistic model-based method for hyperparameter tuning (Gao et al., 2021; Shahriari et al., 2015). For the latest advancements, see Wang et al. (2023) and Yang et al. (2024).

BO uses a surrogate model, often a Gaussian Processes (GP), to approximate the true objective function. Hyperparameters are collected in $\vartheta = [\alpha, \lambda, r]$ and the objective function $\iota[\vartheta]$ is modeled as $\iota[\vartheta] \sim \mathcal{GP}(m[\vartheta], k[\vartheta, \vartheta'])$, where $m[\vartheta]$ is its mean and $k[\vartheta, \vartheta']$ the kernel (variance) function. The objective function is evaluated at $j$ sequential points $\text{MSE}^{(j)} = \iota(\vartheta^{(j)})$, with $\text{MSE}^{(j)} \sim N(\iota(\vartheta^{(j)}), \sigma^2)$. This process induces a posterior over the acquisition function, guiding the selection of the next hyperparameters. Common acquisition functions include probability of improvement (PI), expected improvement (EI), upper confidence bound (UCB), and mutual information (MI) (Snoek et al., 2012). BO starts with an initial set of hyperparameters and objective function values to train the surrogate model. The acquisition function balances the posterior mean ($\varpi(\vartheta)$) for exploitation and variance ($\upsilon(\vartheta)$) for exploration. The GP-UCB is given by

$$\vartheta^{(j+1)} = \underset{\vartheta}{\operatorname{argmax}}\{\varpi(\vartheta) + \varkappa\upsilon(\vartheta)\},$$

where $\varpi(\vartheta)$ is driven by the mean function $m(\vartheta)$, $\upsilon(\vartheta)$ by the variance function $k(\vartheta)$, and $\varkappa$ determines the trade-off between exploitation and exploration. Contal et al. (2014) improved GP-UCB with the Gaussian Process Mutual Information algorithm (GP-MI) as $\vartheta^{(j+1)} = \underset{\vartheta}{\operatorname{argmax}}\{\mu(\vartheta^{(j)}) + \sqrt{\log(2/\varrho)}(\sqrt{\Sigma(\vartheta^{(j)}) + \varsigma^{(j-1)}} - \sqrt{\varsigma^{(j-1)}})\}$, where $\varsigma$ controls exploration, $0 < \varrho < 1$, and $\Sigma(\vartheta^{(j)})$ is the variance function at $\vartheta^{(j)}$.

## 6 Numerical Experiments

### 6.1 Materials

We evaluate our proposed method using one simulated genomic dataset as well as two real-world genomic datasets. Specifically, single-nucleotide polymorphisms (SNPs), which is a type of genetic variation that represent differences in one of the two nucleotides that make up an individual's DNA at a specific location compared to the most common nucleotide pair found in a population. SNPs are typically represented by a count of 0, 1, or 2, where 0 means both nucleotides at the SNP location match the most common pair, 1 indicates that one of the two nucleotides differs from the common pair and 2 means both nucleotides at the location differ from the most common pair. This count system helps quantify genetic variation at a specific SNP site. A detailed explanation of these datasets is provided in Appendix C.

Simulated QTLMAS 2010 Dataset (Szydlowski & Paczyńska, 2011): This dataset comprises 3226 individuals, with genomic single nucleotide polymorphism (SNP) data organized in a matrix $X$ of size $3226 \times 9723$, with two observed traits (response variables): a quantitative and a binary trait. In this study, the quantitative trait was chosen as the phenotype which is represented as a continuous response vector ($y$) of length 3226.

Real Pig Dataset (Cleveland et al., 2012): This dataset contains genomic SNP data from 3534 individuals, organized in a matrix of size $3534 \times 52842$, along with phenotypic data for five traits. We used trait 4, which had a heritability of 0.58, as the phenotype that constitutes a vector ($y$) of length 3534.

Real Mice Dataset (Pérez & de Los Campos, 2014): This dataset contains data from 1814 individuals, with a genomic SNP data matrix of size $1814 \times 10346$, along with two traits: body length (BL) and body mass index (BMI). In this study, the continuous trait BL was chosen to be our response vector ($y$) that has a length of 1814.

### 6.2 Results

The WL1L0-ADMM, WL1L0-SCPRSM, EN-ADMM, EN-SCPRSM, LASSO-ADMM and LASSO-SCPRSM methods were implemented in Julia 1.10.1 (Bezanson et al., 2017) using the ProximalOperators package (Antonello et al., 2018). For SCAD-ADMM, SCAD-SCPRSM, MCP-ADMM and MCP-SCPRSM, we wrote our own code manually in Julia. For all methods, the BO was performed with the BayesianOptimization package using an ElasticGPE model and the squared exponential automatic relevance determination (SEArd) kernel (Fairbrother et al., 2018). The initial values of $\hat{b}$, $\hat{c}$ and $\hat{d}$ were set to the marginal covariances between $y$ and $X$, multiplied by 0.0001. By conducting preliminary runs for each set of hyperparameters using BO, we identified the optimal range of parameters. BO with the MI acquisition function was executed for hyperparameter tuning of all methods. The regression coefficients of the model are obtained from the training dataset, and once the model is trained, it predicts outcomes on the test dataset. The MSE is then calculated on the test dataset to assess the model's generalization performance. The test MSE was monitored during the BO process to ensure convergence, which was indicated by no further decrease in MSE. All analyses were executed on a Linux computing platform equipped with an AMD EPYC 7302P 16-Core Processor and 32GB of system memory.

#### 6.2.1 Simulated QTLMAS 2010 Dataset

BO was executed for 250 iterations with 4 GP function evaluations per iteration across all methods. The lower and upper bounds for $\lambda_1$ were set to 0.001 and 1000.0, 0.1 and 1800.0, and 0.001 and 600.0 for LASSO-ADMM, SCAD-ADMM, and MCP-ADMM, respectively. The lower and upper bounds for $r$ were set to 0.01 and 0.999, 0.001 and 1.0, 0.001 and 1.0, and for $\lambda_1$ were set to 0.001 and 1000.0, 0.1 and 1800.0, and 0.001 and 600.0 for LASSO-SCPRSM, SCAD-SCPRSM, and MCP-SCPRSM, respectively. For EN-ADMM the lower and upper bounds for $\lambda_1$ were set to 10.0 and 600.0 and for $\lambda_2$, they were set to 0.001 and 1.0, respectively. For EN-SCPRSM the lower and upper bounds for $\lambda_1$ were set to 10.0 and 500.0, for $\lambda_2$, they were set to 0.001 and 200.0, for $r$, they were set to 0.01 and 0.99 respectively. For WL1L0-ADMM the lower and upper bounds for $\alpha$ were set to 0.01 and 0.99, and for $\lambda_1$, they were set to 0.001 and 500.0, respectively. For WL1L0-SCPRSM, the lower and upper bounds for $\alpha$ were set to 0.0001 and 0.999, for $r$ they were set to 0.0001 and 1.0, and for $\lambda_1$ they were set to 0.0001 and 500.0, respectively. The best result, with a minimum test MSE of 64.55, was found with WL1L0-SCPRSM at $\lambda_1 = 391.55$, $\alpha = 0.90$, and $r = 0.48$ (Table 1). Timing of the last evaluation with optimized parameters showed that SCAD-ADMM executed most quickly in only 6.68 seconds. It should be noted that those methods with one regularization parameter tend to be faster to train compared to other methods with two or three hyperparameters.

#### 6.2.2 Real Pig Dataset

For the Pig dataset, we employed 5-fold cross-validation with random allocations into training and test data to obtain the minimum test MSE on the test data set, with the results averaged over the folds. Here, for all methods, BO was executed for 100 iterations with 3 GP function evaluations per iteration due to the large dataset size. The lower and upper bounds for $\lambda_1$ were set to 50.0 and 600.0, 0.1 and 1000.0, and 0.1 and 20.0 for LASSO-ADMM, SCAD-ADMM, and MCP-ADMM, respectively. The lower and upper bounds for

| Method | min MSE | $\lambda_1$ | $\lambda_2$ | $\alpha$ | r | Time (s) | Number of non-zeros |
|---|---|---|---|---|---|---|---|
| LASSO-ADMM | 66.55 | 294.22 | - | - | - | 9.31 | 417 |
| LASSO-SCPRSM | 65.95 | 312.51 | - | | 0.19 | 21.51 | 334 |
| SCAD-ADMM | 66.50 | 309.65 | - | - | - | 6.68 | 386 |
| SCAD-SCPRSM | 65.91 | 299.79 | - | - | 0.42 | 19.68 | 352 |
| MCP-ADMM | 69.89 | 362.57 | - | - | - | 10.10 | 3898 |
| MCP-SCPRSM | 68.12 | 268.17 | - | - | 0.99 | 17.93 | 3883 |
| EN-ADMM | 66.52 | 307.65 | 0.79 | - | - | 11.87 | 390 |
| EN-SCPRSM | 65.92 | 288.15 | 0.001 | - | 0.99 | 22.18 | 375 |
| WL1L0-ADMM | 64.77 | 370.46 | - | 0.86 | - | 28.63 | 324 |
| WL1L0-SCPRSM | **64.55** | 391.55 | - | 0.90 | 0.48 | 25.39 | **275** |

Table 1: Performance evaluation of various regularization methods with optimal parameters on simulated QTLMAS data. The best-performing test MSE and most sparse model are highlighted in bold.

$\lambda_1$ were set to 50.0 and 400.0, 50.0 and 400.0, and 0.01 and 20.0, and lower and upper bounds for $r$ were set to 0.01 and 1.0, 0.01 and 1.0, and 0.01 and 1.0 for LASSO-SCPRSM, SCAD-SCPRSM, and MCP-SCPRSM, respectively.

For EN-ADMM the lower and upper bounds for $\lambda_1$ were set to 10.0 and 600.0 and for $\lambda_2$, they were set to 0.001 and 1.0 respectively. For EN-SCPRSM the lower and upper bounds for $\lambda_1$ were set to 0.1 and 200.0, for $\lambda_2$, they were set to 0.01 and 100.0, for $r$, they were set to 0.001 and 1.0, respectively. For WL1L0-ADMM the lower and upper bounds for $\alpha$ were set to 0.001 and 0.99, and for $\lambda_1$, they were set to 0.001 and 100.0, respectively. For WL1L0-SCPRSM, the lower and upper bounds for $\alpha$ were set to 0.001 and 0.99, for $r$ they were set to 0.001 and 1.0, and for $\lambda_1$ they were set to 0.0001 and 200.0, respectively. We observed little variability in the minimum test MSE across the CV-folds for all methods. Hence, we report the mean minimum test MSE using the average estimates of the respective parameters for all methods. The best result, with a mean minimum test MSE of 4.48, was found with WL1L0-SCPRSM with mean estimates $\lambda_1 = 240.63$, $\alpha = 0.54$, and $r = 0.41$ (Table 2). The average timing over the folds of the last evaluation with optimized regularization parameters showed that SCAD-ADMM was fastest, taking only 25.8 seconds.

| Method | min MSE | $\lambda_1$ | $\lambda_2$ | $\alpha$ | r | Time (s) | Number of non-zeros |
|---|---|---|---|---|---|---|---|
| LASSO-ADMM | 4.53 | 118.75 | - | - | - | 26.2 | 1515 |
| LASSO-SCPRSM | 4.50 | 115.63 | - | | 0.32 | 48.21 | 1200 |
| SCAD-ADMM | 4.64 | 127.80 | - | - | - | 25.8 | 1272 |
| SCAD-SCPRSM | 4.50 | 115.63 | - | - | 0.32 | 48.68 | 1200 |
| MCP-ADMM | 6.13 | 100.0 | - | - | - | 27.87 | 22909 |
| MCP-SCPRSM | 6.12 | 106.25 | - | - | 0.94 | 25.58 | 21627 |
| EN-ADMM | 4.53 | 120.63 | 0.31 | - | - | 26.86 | 1447 |
| EN-SCPRSM | 4.51 | 118.79 | 96.88 | - | 0.97 | 32.42 | 1433 |
| WL1L0-ADMM | 4.49 | 43.75 | - | 0.56 | - | 130.82 | 1093 |
| WL1L0-SCPRSM | **4.48** | 240.63 | - | 0.54 | 0.41 | 124.89 | **852** |

Table 2: Evaluation of the performance of various regularization methods with optimal parameters, averaged across five CV-folds on the pig data. The best-performing test MSE and most sparse model are highlighted in bold.

### 6.2.3 Real Mice Dataset

Similar to the Pig dataset, we employed 5-fold cross-validation also for this data. BO was executed for 100 iterations with 4 GP function evaluations per iteration across all methods. The lower and upper bounds for $\lambda_1$ were set to 0.0001 and 20.0, 0.001 and 20.0, and 0.1 and 20.0 for LASSO-ADMM, SCAD-ADMM, and MCP-ADMM, respectively. The lower and upper bounds for $\lambda_1$ were set to 0.001 and 35.0, 0.001 and 35.0, and 0.01 and 20.0, and lower and upper bounds for $r$ were set to 0.001 and 1.0, 0.001 and 1.0, and 0.01 and

1.0 for LASSO-SCPRSM, SCAD-SCPRSM, and MCP-SCPRSM, respectively. For EN-ADMM the lower and upper bounds for $\lambda_1$ were set to 0.01 and 18.0 and for $\lambda_2$, they were set to 0.001 and 1.0 respectively. For EN-SCPRSM the lower and upper bounds for $\lambda_1$ were set to 0.1 and 42.0, for $\lambda_2$, they were set to 0.01 and 40.0, for $r$, they were set to 0.01 and 1.0, respectively.

For WL1L0-ADMM, the lower and upper bounds for $\alpha$ were set to 0.001 and 0.99, and for $\lambda_1$, they were set to 0.001 and 100.0, respectively. For WL1L0-SCPRSM, the lower and upper bounds for $\alpha$ were set to 0.001 and 0.99, for $r$ they were set to 0.001 and 1.0, and for $\lambda_1$ they were set to 0.0001 and 200.0, respectively. The best result, with a mean minimum test MSE of 0.259 was found with WL1L0-SCPRSM at the average estimates $\lambda_1 = 56.25$, $\alpha = 0.28$, and $r = 0.16$ (Table 3).

The average timing over the folds of the last evaluation with optimized regularization parameters showed that SCAD-ADMM once again was fastest with a time of only 2.08 seconds.

| Method | min MSE | $\lambda_1$ | $\lambda_1$ | $\alpha$ | r | Time (s) | Number of non-zeros |
|--------|---------|-------------|-------------|----------|---|----------|---------------------|
| LASSO-ADMM | 0.273 | 20.0 | - | - | - | 2.10 | 319 |
| LASSO-SCPRSM | 0.267 | 24.06 | - | | 0.81 | 3.14 | 280 |
| SCAD-ADMM | 0.274 | 24.0 | - | - | - | 2.08 | 319 |
| SCAD-SCPRSM | 0.267 | 24.06 | - | - | 0.81 | 2.78 | 280 |
| MCP-ADMM | 0.273 | 20.0 | - | - | - | 2.14 | 432 |
| MCP-SCPRSM | 0.273 | 20.0 | - | - | 0.01 | 26.62 | 387 |
| EN-ADMM | 0.276 | 18.0 | 0.99 | - | - | 2.15 | 539 |
| EN-SCPRSM | 0.267 | 24.98 | 38.75 | - | 0.97 | 2.51 | 256 |
| WL1L0-ADMM | 0.265 | 43.75 | - | 0.56 | - | 19.97 | 216 |
| WL1L0-SCPRSM | **0.259** | 56.25 | - | 0.28 | 0.16 | 22.83 | **188** |

Table 3: Evaluation of the performance of various regularization methods with optimal parameters, averaged across five CV-folds on the mice dataset. The best-performing test MSE and most sparse model are highlighted in bold.

# 7 Discussion

The WL1L0 method demonstrates superior performance across all datasets by achieving the lowest MSE and the fewest non-zero coefficients. This highlights its effectiveness and efficiency as a regularization technique in high-dimensional data analysis, making it a valuable alternative to the LASSO, SCAD, MCP and EN. The weighting parameter $\alpha$ in WL1L0 provides flexibility in tuning the regularization effect, making the method adaptable to different datasets and problem settings. This adaptability enhances its robustness and applicability across diverse scenarios.

It has been demonstrated several times that the SCAD and MCP often outperform the LASSO (Fan et al., 2014a; Fan & Li, 2001; Zhang, 2010). However, while the LASSO, SCAD, MCP and EN also offer competitive approaches to regularization, the WL1L0 method consistently outperforms them, providing enhanced model sparsity and interpretability without compromising predictive accuracy. The joint sparsity induced by the $L^1$ and the $L^0$ components make the resulting model more interpretable. This is crucial in many scientific and industrial applications, where understanding the model is as important as its predictive power.

The use of the SCPRSM algorithm introduces an additional parameter $r$, which allows for finer control over the optimization process and potentially leads to better convergence properties and more precise model fitting. Across all datasets used, the SCPRSM variants demonstrate strong performance by achieving the smallest minimum MSEs while maintaining a manageable number of non-zero coefficients. Specifically, WL1L0-SCPRSM consistently achieves the lowest MSE across all our datasets, demonstrating its superior ability to minimize prediction errors. It is likely that it will be highly effective in terms of both accuracy and reliability across other types of data. Several other studies have shown that SCPRSM outperforms ADMM (Li & Yuan, 2015; Li et al., 2021).

In this paper, we have mostly focused on the regularization part and note that there certainly is room for computational advancements. ADMM can be improved using techniques such as accelerated ADMM (Zhang et al., 2019; Zeng et al., 2024, and references therein) as well as stochastic distributed ADMM (Chen et al., 2021, and references therein). Similarly, for SCPRSM, further computational improvements can be achieved via stochastic SCPRSM (Na et al., 2017) and indefinite-proximal SCPRSM (Gu et al., 2022; Bai et al., 2023).

## 8 Conclusion

This paper introduces a novel joint weighted $L^1$- and $L^0$-norm method denoted WL1L0 based on proximal mappings and translation functions, aiming to debias the bias introduced by the $L^1$-norm when applied to high-dimensional data. Our model introduces a weighting parameter $\alpha$, allowing for the adjustment of the influence of both regularizers. The convergence of ADMM and SCPRSM for the developed method is shown under reasonable assumptions. All hyper-parameters are optimized using Bayesian optimization. The WL1L0-SCPRSM method consistently achieves the lowest MSE across all datasets when compared to all other tested regularization methods (LASSO, EN, SCAD and MCP). Hence, the WL1L0-SCPRSM's superior performance across different genomic high-dimensional datasets demonstrates its versatility. Our current paper focuses primarily on prediction. In future work, we plan to specifically address the properties of variable selection.

## Acknowledgements

We acknowledge funding from the University of Oulu & the Academy of Finland Profi 326291.

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

## Appendix

## A   Related Problems

Consider the following formulation

$$\hat{\boldsymbol{b}} = \operatorname*{argmin}_{\boldsymbol{b}} F(\boldsymbol{b}) := \Gamma(\boldsymbol{b}) + \lambda(1-\alpha)||\boldsymbol{b}||_0, \tag{35}$$

where $\Gamma(\boldsymbol{b}) := ||\boldsymbol{y} - \boldsymbol{X}\boldsymbol{b}||_2^2 + \lambda\alpha||\boldsymbol{b}||_1$. Since $\Gamma(\boldsymbol{b})$ is a convex function, it has a global minimum value. By the Weierstrass theorem, a continuous function over a nonempty compact set attains a minimum. The existence of an optimal solution is guaranteed if a function is continuous over a closed set and coercive over the set (Bertsekas, 2016). Beck (2017) demonstrates that the latter extends to closed functions, i.e. a closed and coercive function over a closed set attains an optimal solution.

For the case $||\boldsymbol{b}||_0 = \sum_{i=1}^{p} \mathbf{1}(b_i \neq 0)$ with $\lambda(1-\alpha) > 0$, we need to show it is a closed function. Let $g(\boldsymbol{b}) = \sum_{i=1}^{p} I(b_i)$, where $I : \mathbb{R} \to \{0,1\}$ is defined as

$$I(b_i) = \begin{cases} \lambda(1-\alpha), & b_i \neq 0, \\ 0, & b_i = 0. \end{cases}$$

The function $I(.)$ is closed since its level sets, given by

$$\operatorname{Lev}(I, \eta) = \begin{cases} \emptyset, & \eta < 0, \\ \{0\}, & \eta \in [0,1), \\ \mathbb{R}, & \eta \geq 1, \end{cases} \tag{36}$$

are closed sets. Here, $g$ is a closed function. Furthermore, using Theorem 2.6 in (Beck, 2017), the closedness of $||\boldsymbol{b}||_0$ implies its lower semi-continuity.

A vector $\boldsymbol{b}^*$ is a local minimum of the function $F$, if there exists $\varepsilon > 0$ such that $F(\boldsymbol{b}^*) \leq F(\boldsymbol{b})$ for all $\boldsymbol{b} \in \mathbb{R}^p$ with $||\boldsymbol{b} - \boldsymbol{b}^*|| < \varepsilon$. A vector $\boldsymbol{b}^*$ is a global minimum if $F(\boldsymbol{b}^*) \leq F(\boldsymbol{b})$ for all $\boldsymbol{b} \in \mathbb{R}^p$.

For illustration purposes, we generated a random design matrix $\boldsymbol{X}$ with dimension $100 \times 500$ by simulating 100 samples and 500 features. For each value in the range between -5 and 5, the outcomes of a function $F(b)$ that combines the squared error loss with the regularization term that consists of the weighted sum of the $L^1$ and $L^0$-norms were produced. The regularization parameters were set to $\lambda = 2$ and $\alpha = 0.6$. Therefore, the plotted $F(b)$ includes both the error and regularization terms, rather than solely the penalty norms. One can see that $F(b)$ is nonconvex because any point between the endpoints $A$ and $B$, as indicated by the dashed red line in Figure 1, lies outside the domain of $F(b)$. In fact, the shape of the function $F(b)$ is similar to that of nonconvex regularization methods such as SCAD and MCP (Fan & Li, 2001; Zhang, 2010; Zhao et al., 2018).

Various shapes of the function $F(b)$ for different values of $\alpha$ and $\lambda = 1$ are depicted in Figure 2 to illustrate the impact of $\alpha$ on the function's behavior.

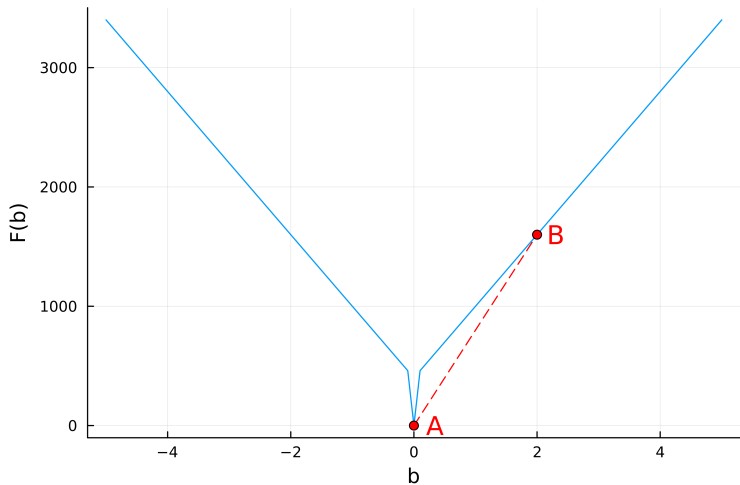

Figure 1: Illustration of the nonconvexity of the function $F$

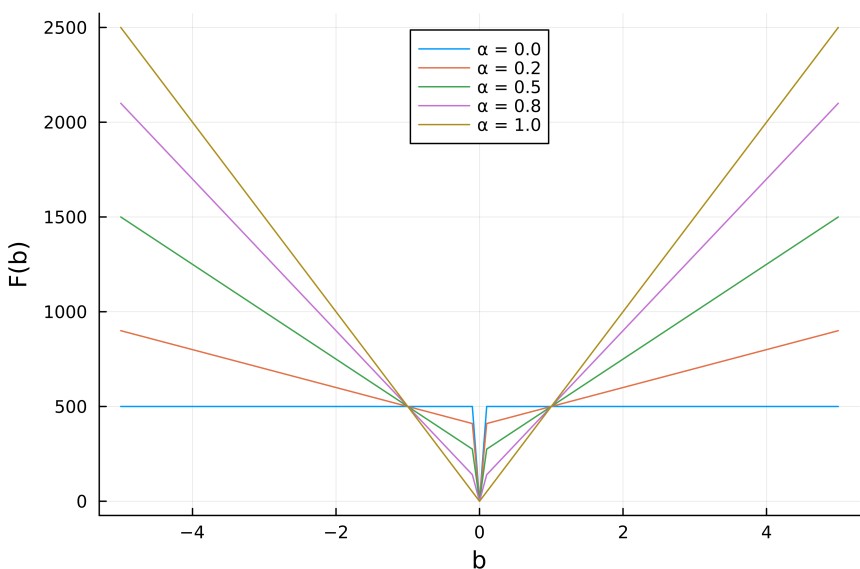

Figure 2: Different function values of $F$ with regularizer $\lambda = 1$ and $\alpha = 0$ $(L^0), 0.2, 0.5, 0.8, 1$ $(L^1)$.

# B Proofs

## B.1 Proof of Lemma 1

Given $f(\boldsymbol{u}) = \langle \boldsymbol{m}, \boldsymbol{u} \rangle + a$, the proximal operator of the function $f(\boldsymbol{u})$ is defined as $\text{prox}_f(\boldsymbol{u}) = \underset{\boldsymbol{v}}{\arg\min} \left\{ f(\boldsymbol{v}) + \frac{1}{2} \|\boldsymbol{v} - \boldsymbol{u}\|_2^2 \right\}$. Substituting the affine function $f(\boldsymbol{v}) = \langle \boldsymbol{m}, \boldsymbol{v} \rangle + a$ gives us $\text{prox}_f(\boldsymbol{u}) = \underset{\boldsymbol{v}}{\arg\min} \left\{ \langle \boldsymbol{m}, \boldsymbol{v} \rangle + a + \frac{1}{2} \|\boldsymbol{v} - \boldsymbol{u}\|_2^2 \right\}$. The term $a$ is constant and does not affect the minimization (it doesn't depend on $\boldsymbol{v}$), so we can ignore it. Next, the squared norm term $\frac{1}{2} \|\boldsymbol{v} - \boldsymbol{u}\|_2^2$ can be expanded as $\frac{1}{2} \|\boldsymbol{v} - \boldsymbol{u}\|_2^2 = \frac{1}{2} \left( \|\boldsymbol{v}\|_2^2 - 2\langle \boldsymbol{v}, \boldsymbol{u} \rangle + \|\boldsymbol{u}\|_2^2 \right)$. Since $\|\boldsymbol{u}\|_2^2$ is constant with respect to $\boldsymbol{v}$, it can also be ignored in the minimization. Now, combining the linear terms involving $\boldsymbol{v}$, we have $\underset{\boldsymbol{v}}{\arg\min} \left\{ \frac{1}{2} \|\boldsymbol{v}\|_2^2 + \langle \boldsymbol{m} - \boldsymbol{u}, \boldsymbol{v} \rangle \right\}$. This expression is a quadratic function in $\boldsymbol{v}$. To minimize it, we set the gradient of the objective function

with respect to $\boldsymbol{v}$ equal to zero. The gradient of $\frac{1}{2}\|\boldsymbol{v}\|_2^2$ is $\boldsymbol{v}$. The gradient of $\langle \boldsymbol{m} - \boldsymbol{u}, \boldsymbol{v} \rangle$ is $\boldsymbol{m} - \boldsymbol{u}$. Setting the total gradient to zero gives us: $\boldsymbol{v} + (\boldsymbol{m} - \boldsymbol{u}) = 0$. Solving for $\boldsymbol{v}$, we find $\boldsymbol{v} = \boldsymbol{u} - \boldsymbol{m}$. Therefore, $\text{prox}_f(\boldsymbol{u}) = \boldsymbol{u} - \boldsymbol{m}$.

## B.2 Proof of Theorem 1

While our model differs from that of Wang et al. (2019; 2018), we adopt a similar proof framework.

**Proof (a):** From (23a), $\boldsymbol{b}^{(k+1)}$ minimizes $L_\gamma(\boldsymbol{b}^{(k)}, \boldsymbol{u}^{(k)}, \boldsymbol{m}^{(k)})$ and since $L_\gamma(\boldsymbol{b}, \boldsymbol{u}, \boldsymbol{m})$ is strongly convex with respect to $\boldsymbol{b}$, the Lagrangian function satisfies the following inequality (Beck, 2017):

$$L_\gamma(\boldsymbol{b}^{(k+1)}, \boldsymbol{u}^{(k)}, \boldsymbol{m}^{(k)}) - L_\gamma(\boldsymbol{b}^{(k)}, \boldsymbol{u}^{(k)}, \boldsymbol{m}^{(k)}) \leq -\frac{\rho}{2}\|\boldsymbol{b}^{(k+1)} - \boldsymbol{b}^{(k)}\|_2^2, \tag{37}$$

where $L_\gamma(\cdot)$ is a $\rho$-strongly convex function ($\rho > 0$). From the augmented Lagrangian function in (20), we have

$$L_\gamma(\boldsymbol{b}^{(k+1)}, \boldsymbol{u}^{(k+1)}, \boldsymbol{m}^{(k+1)}) - L_\gamma(\boldsymbol{b}^{(k+1)}, \boldsymbol{u}^{(k+1)}, \boldsymbol{m}^{(k)}) = \gamma \left(\boldsymbol{b}^{(k+1)} - \boldsymbol{u}^{(k+1)}\right)^{\mathsf{T}} \left(\boldsymbol{m}^{(k+1)} - \boldsymbol{m}^{(k)}\right). \tag{38}$$

Now we rewrite (23c) as

$$\boldsymbol{b}^{(k+1)} - \boldsymbol{u}^{(k+1)} = \boldsymbol{m}^{(k+1)} - \boldsymbol{m}^{(k)}. \tag{39}$$

From (23b) and (29), we get

$$\nabla f(\boldsymbol{b}^{(k+1)}) + \gamma(\boldsymbol{b}^{(k+1)} - \boldsymbol{u}^{(k+1)} + \boldsymbol{m}^{(k)}) = 0. \tag{40}$$

Substituting (23c) into (40), we obtain

$$\nabla f(\boldsymbol{b}^{(k+1)}) = -\gamma \boldsymbol{m}^{(k+1)}. \tag{41}$$

Using (39) and (41), (38) becomes

$$\gamma \left(\|\boldsymbol{m}^{(k+1)} - \boldsymbol{m}^{(k)}\|^2\right) = \gamma \left(\| -\frac{1}{\gamma}\nabla f(\boldsymbol{b}^{(k+1)}) + \frac{1}{\gamma}\nabla f(\boldsymbol{b}^{(k)})\|^2\right) \leq \frac{l_f^2}{\gamma} \left(\|\boldsymbol{b}^{(k+1)} - \boldsymbol{b}^{(k)}\|^2\right). \tag{42}$$

Hence,

$$L_\gamma(\boldsymbol{b}^{(k+1)}, \boldsymbol{u}^{(k+1)}, \boldsymbol{m}^{(k+1)}) - L_\gamma(\boldsymbol{b}^{(k+1)}, \boldsymbol{u}^{(k+1)}, \boldsymbol{m}^{(k)}) \leq \frac{l_f^2}{\gamma} \left(\|\boldsymbol{b}^{(k+1)} - \boldsymbol{b}^{(k)}\|^2\right). \tag{43}$$

Here, the term $l_f \geq 0$ denotes a Lipschitz gradient of the function $f(\boldsymbol{b})$. From (23b) we have that

$$L_\gamma(\boldsymbol{b}^{(k+1)}, \boldsymbol{u}^{(k+1)}, \boldsymbol{m}^{(k)}) - L_\gamma(\boldsymbol{b}^{(k+1)}, \boldsymbol{u}^{(k)}, \boldsymbol{m}^{(k)}) \leq 0. \tag{44}$$

Finally, combining (37), (43) and (44), we obtain the desired inequality as

$$\begin{aligned} L_\gamma(\boldsymbol{b}^{(k+1)}, & \boldsymbol{u}^{(k+1)}, \boldsymbol{m}^{(k+1)}) - L_\gamma(\boldsymbol{b}^{(k)}, \boldsymbol{u}^{(k)}, \boldsymbol{m}^{(k)}) \\ &= L_\gamma(\boldsymbol{b}^{(k+1)}, \boldsymbol{u}^{(k+1)}, \boldsymbol{m}^{(k+1)}) - L_\gamma(\boldsymbol{b}^{(k+1)}, \boldsymbol{u}^{(k+1)}, \boldsymbol{m}^{(k)}) \\ &\quad + L_\gamma(\boldsymbol{b}^{(k+1)}, \boldsymbol{u}^{(k+1)}, \boldsymbol{m}^{(k)}) - L_\gamma(\boldsymbol{b}^{(k+1)}, \boldsymbol{u}^{(k)}, \boldsymbol{m}^{(k)}) \\ &\quad + L_\gamma(\boldsymbol{b}^{(k+1)}, \boldsymbol{u}^{(k)}, \boldsymbol{m}^{(k)}) - L_\gamma(\boldsymbol{b}^{(k)}, \boldsymbol{u}^{(k)}, \boldsymbol{m}^{(k)}) \\ &\leq \left(\frac{l_f^2}{\gamma} - \frac{\rho}{2}\right) \|\boldsymbol{b}^{(k+1)} - \boldsymbol{b}^{(k)}\|_2^2 \\ &= -\delta_1 \|\boldsymbol{b}^{(k+1)} - \boldsymbol{b}^{(k)}\|_2^2, \end{aligned} \tag{45}$$

where $\sigma_1 = \frac{\rho}{2} - \frac{l_f^2}{\gamma}$ and $\gamma > \frac{2l_f^2}{\rho}$. Hence, the sufficient decreasing condition is met.

**Proof (b):** We utilize the descent lemma to prove that $L_\gamma(\boldsymbol{b}^{(k)}, \boldsymbol{u}^{(k)}, \boldsymbol{m}^{(k)})$ is lower bounded for any $k$.

**Lemma 2** *(Descent lemma) Let the function $f$ belong to the class of continuously differentiable functions with constant $l_f$ Lipschitz continuous gradients. Then for any two points $\boldsymbol{b}^{(k)}$ and $\boldsymbol{u}^{(k)}$,*

$$f(\boldsymbol{u}^{(k)}) \leq f(\boldsymbol{b}^{(k)}) + \nabla f(\boldsymbol{b}^{(k)})^\top (\boldsymbol{u}^{(k)} - \boldsymbol{b}^{(k)}) + \frac{l_f}{2}||\boldsymbol{u}^{(k)} - \boldsymbol{b}^{(k)}||_2^2. \tag{46}$$

The proof of the descent lemma can be found in (Beck, 2017), see Lemma 5.7.

As a result of the Descent lemma, the sequence is lower bounded as

$$
\begin{aligned}
L_\gamma(\boldsymbol{b}^{(k)}, \boldsymbol{u}^{(k)}, \boldsymbol{m}^{(k)}) =& f(\boldsymbol{b}) + g(\boldsymbol{u}) + \frac{\gamma}{2}||\boldsymbol{b} - \boldsymbol{u} + \frac{1}{\gamma}\boldsymbol{m}||_2^2 - \frac{\gamma}{2}||\frac{1}{\gamma}\boldsymbol{m}||_2^2 \\
=& f(\boldsymbol{b}^{(k)}) + g(\boldsymbol{u}^{(k)}) + \boldsymbol{m}^T(\boldsymbol{b}^{(k)} - \boldsymbol{u}^{(k)}) + (\gamma/2)||\boldsymbol{b}^{(k)} - \boldsymbol{u}^{(k)}||_2^2 \\
\geq& f(\boldsymbol{u}^{(k)}) + g(\boldsymbol{u}^{(k)}) + \left(\frac{\gamma}{2} - \frac{l_f}{2}\right)||\boldsymbol{u}^{(k)} - \boldsymbol{m}^{(k)}||_2^2 \\
\geq& -\infty \text{ for } \gamma \geq l_f.
\end{aligned} \tag{47}
$$

Hence, from (47), $L_\gamma(\boldsymbol{b}^{(k)}, \boldsymbol{u}^{(k)}, \boldsymbol{m}^{(k)})$ is lower bounded.

As established in the proof (a), the sufficient descent property implies that $L_\gamma(\boldsymbol{b}^{(k)}, \boldsymbol{u}^{(k)}, \boldsymbol{m}^{(k)})$ is upper-bounded by $L_\gamma(\boldsymbol{b}^0, \boldsymbol{u}^0, \boldsymbol{m}^0)$. To prove that the sequence $\{\boldsymbol{b}^{(k)}, \boldsymbol{u}^{(k)}, \boldsymbol{m}^{(k)}\}$ is bounded, we start by rewriting (45) as

$$
\begin{aligned}
||\boldsymbol{b}^{(k+1)} - \boldsymbol{b}^{(k)}||_2^2 \leq& \frac{1}{\delta_1}(L_\gamma(\boldsymbol{b}^{(k)}, \boldsymbol{u}^{(k)}, \boldsymbol{m}^{(k)}) - L_\gamma(\boldsymbol{b}^{(k+1)}, \boldsymbol{u}^{(k+1)}, \boldsymbol{m}^{(k+1)})) \\
\sum_{k=0}^{l}||\boldsymbol{b}^{(k+1)} - \boldsymbol{b}^{(k)}||_2^2 \leq& \frac{1}{\delta_1}(L_\gamma(\boldsymbol{b}^0, \boldsymbol{u}^0, \boldsymbol{m}^0) - L_\gamma(\boldsymbol{b}^{l+1}, \boldsymbol{u}^{l+1}, \boldsymbol{m}^{l+1})) \\
<& \infty.
\end{aligned} \tag{48}
$$

Equation (48) also holds as $l \to \infty$. Hence, $\boldsymbol{b}^{(k)}$ is bounded.

From (42), we obtain

$$||\boldsymbol{m}^{(k+1)} - \boldsymbol{m}^{(k)}||_2^2 \leq \frac{l_f^2}{\gamma^2}||\boldsymbol{b}^{(k+1)} - \boldsymbol{b}^{(k)}||_2^2.$$

$$\sum_{k=0}^{l}||\boldsymbol{m}^{(k+1)} - \boldsymbol{m}^{(k)}||_2^2 < \infty. \tag{49}$$

This implies that $\boldsymbol{m}^{(k)}$ is bounded.

Finally, from (39) we obtain $\boldsymbol{u}^{(k+1)} = \boldsymbol{b}^{(k+1)} - \boldsymbol{m}^{(k+1)} + \boldsymbol{m}^{(k)}$ and $\boldsymbol{u}^{(k)} = \boldsymbol{b}^{(k)} - \boldsymbol{m}^{(k)} + \boldsymbol{m}^{k-1}$. Then

$$
\begin{aligned}
||\boldsymbol{u}^{(k+1)} - \boldsymbol{u}^{(k)}||_2^2 =& ||\boldsymbol{b}^{(k+1)} - \boldsymbol{b}^{(k)} + \boldsymbol{m}^{(k)} - \boldsymbol{m}^{(k+1)} + \boldsymbol{m}^{k-1} - \boldsymbol{m}^{(k)}||_2^2 \\
\leq& ||\boldsymbol{b}^{(k+1)} - \boldsymbol{b}^{(k)}||_2^2 + ||\boldsymbol{m}^{(k+1)} - \boldsymbol{m}^{(k)}||_2^2 + ||\boldsymbol{m}^{(k)} - \boldsymbol{m}^{k-1}||_2^2.
\end{aligned}
$$

Consequently, we obtain

$$\sum_{k=1}^{\infty}||\boldsymbol{u}^{(k+1)} - \boldsymbol{u}^{(k)}||_2^2 < \infty. \tag{50}$$

Hence, the sequence $\{\boldsymbol{b}^{(k)}, \boldsymbol{u}^{(k)}, \boldsymbol{m}^{(k)}\}$ is bounded.

**Proof (c):** The augmented Lagrangian function $L_\gamma(\boldsymbol{b}, \boldsymbol{u}, \boldsymbol{m}) = f(\boldsymbol{b}) + g(\boldsymbol{u}) + \frac{\gamma}{2}||\boldsymbol{b} - \boldsymbol{u} + \boldsymbol{m}||_2^2 - \frac{\gamma}{2}||\boldsymbol{m}||_2^2$ defined as $L_\gamma : \mathbb{R}^n \to (-\infty, \infty]$ is proper and lower semi-continuous, where $f(\boldsymbol{b}) = ||\boldsymbol{y} - \boldsymbol{X}\boldsymbol{b}||_2^2$, $g(\boldsymbol{u}) = ||\boldsymbol{u}||_1 + ||\boldsymbol{u}||_0$ and $h(\boldsymbol{b}, \boldsymbol{u}, \boldsymbol{m}) = \frac{\gamma}{2}||\boldsymbol{b} - \boldsymbol{u} + \boldsymbol{m}||_2^2 - \frac{\gamma}{2}||\boldsymbol{m}||_2^2$. If $L_\gamma(\boldsymbol{b}, \boldsymbol{u}, \boldsymbol{m})$ is semi-algebraic, then it satisfies the KŁ property

at any point of its domain. Note that both $f$ and $h$ are real polynomial functions, which are semi-algebraic functions (Attouch et al., 2013; Bolte et al., 2014). Both $||\boldsymbol{u}||_0$ and $||\boldsymbol{u}||_1$ have piecewise linear graphs and are therefore semi-algebraic (see Example 3 and 4 in (Bolte et al., 2014), respectively).

Furthermore, consider that $g_1(\boldsymbol{u}) = \lambda\alpha||\boldsymbol{u}||_1$ and $g_2 = \lambda(1-\alpha)||\boldsymbol{u}||_0$. Their proximal operators have piecewise linear graphs and are perfectly known objects (Attouch et al., 2013; Beck, 2017). The proximal operator for $g_1(\boldsymbol{u}) = \lambda\alpha||\boldsymbol{u}||_1$, $\text{prox}_{g_{1(\lambda\alpha)}}(\boldsymbol{u}) = [|\boldsymbol{u}| - \lambda\alpha]_+ \text{sgn}(\boldsymbol{u})$ (the so-called soft thresholding function) is defined as

$$[|\boldsymbol{u}| - \lambda\alpha]_+ \text{sgn}(\boldsymbol{u}) = \begin{cases} \boldsymbol{u} - \lambda\alpha, & \text{if } \boldsymbol{u} \geq \lambda\alpha, \\ 0, & \text{if } |\boldsymbol{u}| < \lambda\alpha, \\ \boldsymbol{u} + \lambda\alpha, & \text{if } \boldsymbol{u} \leq -\lambda\alpha. \end{cases}$$

Hence, $\text{prox}_{g_{1(\lambda\alpha)}}(\boldsymbol{u})$ has a piecewise-linear graph and is semi-algebraic. The proximal operator for $g_2$ can be written as

$$\text{prox}_{g_2(\lambda(1-\alpha))}(\boldsymbol{u}) = \begin{cases} 0, & \text{if } |\boldsymbol{u}| < \sqrt{2\lambda(1-\alpha)}, \\ \boldsymbol{u}, & \text{if } |\boldsymbol{u}| > \sqrt{2\lambda(1-\alpha)}, \\ \{0, \boldsymbol{u}\}, & \text{if } |\boldsymbol{u}| = \sqrt{2\lambda(1-\alpha)}. \end{cases}$$

Clearly, $\text{prox}_{g_2(\lambda(1-\alpha))}(\boldsymbol{u})$ is also piecewise linear and semi-algebraic. Note that $\text{prox}_{g_2(\lambda(1-\alpha))}(\boldsymbol{u}) = \mathcal{H}_\nu(\boldsymbol{u})$ the so-called hard thresholding operator, is defined as

$$\mathcal{H}_\nu(\boldsymbol{u}) \equiv \begin{cases} 0, & \text{if } |\boldsymbol{u}| < \nu, \\ \boldsymbol{u}, & \text{if } |\boldsymbol{u}| > \nu, \\ \{0, \boldsymbol{u}\}, & \text{if } |\boldsymbol{u}| = \nu, \end{cases}$$

where $\nu = \sqrt{2\lambda(1-\alpha)}$. Here, $g_1 + g_2$ is also semi-algebraic.

Consequently, for any nonnegative real numbers $\lambda$ and $\alpha$, the function $f(\boldsymbol{b}) + \lambda\alpha||\boldsymbol{u}||_1 + \lambda(1-\alpha)||\boldsymbol{u}||_0 + \frac{\gamma}{2}||\boldsymbol{b} - \boldsymbol{u} + \boldsymbol{m}||_2^2 - \frac{\gamma}{2}||\boldsymbol{m}||_2^2$ is semi-algebraic. Hence, we conclude that the Lagrangian function in (20) is a KŁ function.

Since $\{\boldsymbol{b}^{(k)}, \boldsymbol{u}^{(k)}, \boldsymbol{m}^{(k)}\}$ is bounded, there exists a subsequence $\{\boldsymbol{b}^{kl}, \boldsymbol{u}^{kl}, \boldsymbol{m}^{kl}\}$ converging to a stationary point $\{\boldsymbol{b}^*, \boldsymbol{u}^*, \boldsymbol{m}^*\}$, where $l \in \mathbb{N}$. Since the Lagrangian function in (20) is a KŁ function (using the lower semicontinuous property), we have

$$L_\gamma(\boldsymbol{b}^*, \boldsymbol{u}^*, \boldsymbol{m}^*) \leq \lim_{l\to\infty} L_\gamma(\boldsymbol{b}^{kl}, \boldsymbol{u}^{kl}, \boldsymbol{m}^{kl}). \tag{51}$$

In conclusion, all the conditions (a)-(c) in Theorem 1 hold.

### B.3 Proof of Theorem 2

Starting with $r = 1$ (PRSM), we update $\boldsymbol{b}$, $\boldsymbol{m}$, and $\boldsymbol{u}$ iteratively according to (24a - 24d)

$$\boldsymbol{b}^{(k+1)} := \underset{\boldsymbol{b}}{\arg\min} \, L_\gamma(\boldsymbol{b}^{(k)}, \boldsymbol{u}^{(k)}, \boldsymbol{m}^{(k)}), \tag{52a}$$

$$\boldsymbol{m}^{(k+\frac{1}{2})} := \boldsymbol{m}^{(k)} + \boldsymbol{b}^{(k+1)} - \boldsymbol{u}^{(k)}, \tag{52b}$$

$$\boldsymbol{u}^{(k+1)} := \underset{\boldsymbol{u}}{\arg\min} \, L_\gamma(\boldsymbol{b}^{(k+1)}, \boldsymbol{u}^{(k)}, \boldsymbol{m}^{(k+\frac{1}{2})}), \tag{52c}$$

$$\boldsymbol{m}^{(k+1)} := \boldsymbol{m}^{(k+\frac{1}{2})} + \boldsymbol{b}^{(k+1)} - \boldsymbol{u}^{(k+1)}. \tag{52d}$$

**Proof (a):** From (52a), since $\boldsymbol{b}^{(k+1)}$ minimizes $L_\gamma(\boldsymbol{b}^{(k)}, \boldsymbol{u}^{(k)}, \boldsymbol{m}^{(k)})$ and Lagrangian is strongly convex with respect to the variable $\boldsymbol{b}$, (37) holds.

Next, using the augmented Lagrangian function in (20), we compute

$$L_\gamma(\boldsymbol{b}^{(k+1)}, \boldsymbol{u}^{(k+1)}, \boldsymbol{m}^{(k+1)}) - L_\gamma(\boldsymbol{b}^{(k+1)}, \boldsymbol{u}^{(k+1)}, \boldsymbol{m}^{k+\frac{1}{2}}) = \gamma \left(\boldsymbol{b}^{(k+1)} - \boldsymbol{u}^{(k+1)}\right)^\mathsf{T} \left(\boldsymbol{m}^{(k+1)} - \boldsymbol{m}^{k+\frac{1}{2}}\right). \tag{53}$$

Now we rewrite (52d) as

$$\boldsymbol{b}^{(k+1)} - \boldsymbol{u}^{(k+1)} = \boldsymbol{m}^{(k+1)} - \boldsymbol{m}^{k+\frac{1}{2}}. \tag{54}$$

From (52c) and (29), we obtain

$$\nabla f(\boldsymbol{b}^{(k+1)}) + \gamma(\boldsymbol{b}^{(k+1)} - \boldsymbol{u}^{(k+1)} + \boldsymbol{m}^{k+\frac{1}{2}}) = 0. \tag{55}$$

Substituting (52d) into (55), we obtain (41). Again, from (52a) and (29), we obtain

$$\nabla f(\boldsymbol{b}^{(k+1)}) + \gamma(\boldsymbol{b}^{(k+1)} - \boldsymbol{u}^{(k)} + \boldsymbol{m}^{(k)}) = 0. \tag{56}$$

Substituting (52b) into (56), we obtain

$$\nabla f(\boldsymbol{b}^{(k+1)}) = -\gamma \boldsymbol{m}^{k+\frac{1}{2}}. \tag{57}$$

Using (41),(54) and (57), (53) becomes

$$\gamma\left(||\boldsymbol{m}^{(k+1)} - \boldsymbol{m}^{k+\frac{1}{2}}||^2\right) = \gamma\left(|| - \frac{1}{\gamma}\nabla f(\boldsymbol{b}^{(k+1)}) + \frac{1}{\gamma}\nabla f(\boldsymbol{b}^{(k+1)})||^2\right) = 0. \tag{58}$$

Hence,

$$L_\gamma(\boldsymbol{b}^{(k+1)}, \boldsymbol{u}^{(k+1)}, \boldsymbol{m}^{(k+1)}) - L_\gamma(\boldsymbol{b}^{(k+1)}, \boldsymbol{u}^{(k+1)}, \boldsymbol{m}^{k+\frac{1}{2}}) \le 0. \tag{59}$$

Using (20), we have

$$L_\gamma(\boldsymbol{b}^{(k+1)}, \boldsymbol{u}^{(k)}, \boldsymbol{m}^{k+\frac{1}{2}}) - L_\gamma(\boldsymbol{b}^{(k+1)}, \boldsymbol{u}^{(k)}, \boldsymbol{m}^{(k)}) = \gamma\left(\boldsymbol{b}^{(k+1)} - \boldsymbol{u}^{(k)}\right)^{\mathsf{T}}\left(\boldsymbol{m}^{k+\frac{1}{2}} - \boldsymbol{m}^{(k)}\right). \tag{60}$$

Next, we reformulate (52b) as

$$\boldsymbol{b}^{(k+1)} - \boldsymbol{u}^{(k)} = \boldsymbol{m}^{k+\frac{1}{2}} - \boldsymbol{m}^{(k)}. \tag{61}$$

Using (57) and (61), (60) becomes

$$\gamma\left(||\boldsymbol{m}^{k+\frac{1}{2}} - \boldsymbol{m}^{(k)}||^2\right) = \gamma\left(|| - \frac{1}{\gamma}\nabla f(\boldsymbol{b}^{(k+1)}) + \frac{1}{\gamma}\nabla f(\boldsymbol{b}^{(k)})||^2\right). \tag{62}$$

Therefore,

$$L_\gamma(\boldsymbol{b}^{(k+1)}, \boldsymbol{u}^{(k)}, \boldsymbol{m}^{k+\frac{1}{2}}) - L_\gamma(\boldsymbol{b}^{(k+1)}, \boldsymbol{u}^{(k)}, \boldsymbol{m}^{(k)}) \le \frac{l_f^2}{\gamma}\left(||\boldsymbol{b}^{(k+1)} - \boldsymbol{b}^{(k)}||^2\right), \tag{63}$$

where $l_f \ge 0$ is a Lipschitz gradient of the function $f(\boldsymbol{b})$. From (52c) we have that

$$L_\gamma(\boldsymbol{b}^{(k+1)}, \boldsymbol{u}^{(k+1)}, \boldsymbol{m}^{k+\frac{1}{2}}) - L_\gamma(\boldsymbol{b}^{(k+1)}, \boldsymbol{u}^{(k)}, \boldsymbol{m}^{k+\frac{1}{2}}) \le 0. \tag{64}$$

Finally, combining (37), (59), (63) and (64), we get the desired inequality as follows

$$\begin{aligned}
&L_\gamma(\boldsymbol{b}^{(k+1)}, \boldsymbol{u}^{(k+1)}, \boldsymbol{m}^{(k+1)}) - L_\gamma(\boldsymbol{b}^{(k)}, \boldsymbol{u}^{(k)}, \boldsymbol{m}^{(k)}) \\
&= L_\gamma(\boldsymbol{b}^{(k+1)}, \boldsymbol{u}^{(k+1)}, \boldsymbol{m}^{(k+1)}) - L_\gamma(\boldsymbol{b}^{(k+1)}, \boldsymbol{u}^{(k+1)}, \boldsymbol{m}^{k+\frac{1}{2}}) \\
&\quad + L_\gamma(\boldsymbol{b}^{(k+1)}, \boldsymbol{u}^{(k+1)}, \boldsymbol{m}^{k+\frac{1}{2}}) - L_\gamma(\boldsymbol{b}^{(k+1)}, \boldsymbol{u}^{(k)}, \boldsymbol{m}^{k+\frac{1}{2}}) \\
&\quad + L_\gamma(\boldsymbol{b}^{(k+1)}, \boldsymbol{u}^{(k)}, \boldsymbol{m}^{k+\frac{1}{2}}) - L_\gamma(\boldsymbol{b}^{(k+1)}, \boldsymbol{u}^{(k)}, \boldsymbol{m}^{(k)}) \\
&\quad + L_\gamma(\boldsymbol{b}^{(k+1)}, \boldsymbol{u}^{(k)}, \boldsymbol{m}^{(k)}) - L_\gamma(\boldsymbol{b}^{(k)}, \boldsymbol{u}^{(k)}, \boldsymbol{m}^{(k)}) \\
&\le \left(\frac{l_f^2}{\gamma} - \frac{\rho}{2}\right)||\boldsymbol{b}^{(k+1)} - \boldsymbol{b}^{(k)}||_2^2 \\
&= -\delta_1||\boldsymbol{b}^{(k+1)} - \boldsymbol{b}^{(k)}||_2^2,
\end{aligned}$$

which is the sufficient decreasing condition (45).

**Proof (b):** The difference lies in some steps to show the boundedness of $\boldsymbol{u}^{(k)}$ and $\boldsymbol{m}^{(k)}$. The rest is the same as in the proof of Theorem 1(b). From (54) and (61) we obtain $\boldsymbol{u}^{(k+1)} = \boldsymbol{b}^{(k+1)} - \boldsymbol{m}^{(k+1)} + \boldsymbol{m}^{k+\frac{1}{2}}$ and $\boldsymbol{u}^{(k)} = \boldsymbol{b}^{(k+1)} - \boldsymbol{m}^{k+\frac{1}{2}} + \boldsymbol{m}^{(k)}$, respectively. Then

$$
\begin{aligned}
||\boldsymbol{u}^{(k+1)} - \boldsymbol{u}^{(k)}||_2^2 &= ||\boldsymbol{m}^{k+\frac{1}{2}} - \boldsymbol{m}^{(k+1)} + \boldsymbol{m}^{k+\frac{1}{2}} - \boldsymbol{m}^{(k)}||_2^2 \\
&\leq ||\boldsymbol{m}^{(k+1)} - \boldsymbol{m}^{k+\frac{1}{2}}||_2^2 + ||\boldsymbol{m}^{k+\frac{1}{2}} - \boldsymbol{m}^{(k)}||_2^2,
\end{aligned}
$$

This inequality can be rewritten using (58) and (62) as

$$
||\boldsymbol{u}^{(k+1)} - \boldsymbol{u}^{(k)}||_2^2 \leq \frac{l_f^2}{\gamma^2} ||\boldsymbol{b}^{(k+1)} - \boldsymbol{b}^{(k)}||_2^2.
$$

Consequently, we obtain

$$
\sum_{k=0}^{\infty} ||\boldsymbol{u}^{(k+1)} - \boldsymbol{u}^{(k)}||_2^2 < \infty. \tag{65}
$$

Equation (65) implies that $\boldsymbol{u}^{(k)}$ is bounded. To show that $\boldsymbol{m}^{(k)}$ is bounded, we analyze the difference $\boldsymbol{m}^{(k+1)} - \boldsymbol{m}^{(k)}$ as follows $\boldsymbol{m}^{(k+1)} - \boldsymbol{m}^{(k)} = \boldsymbol{m}^{(k+1)} - \boldsymbol{m}^{k+\frac{1}{2}} + \boldsymbol{m}^{k+\frac{1}{2}} - \boldsymbol{m}^{(k)}$. Then, we obtain

$$
||\boldsymbol{m}^{(k+1)} - \boldsymbol{m}^{(k)}||_2^2 \leq \frac{l_f^2}{\gamma^2} ||\boldsymbol{b}^{(k+1)} - \boldsymbol{b}^{(k)}||_2^2,
$$

$$
\sum_{k=0}^{\infty} ||\boldsymbol{m}^{(k+1)} - \boldsymbol{m}^{(k)}||_2^2 < \infty. \tag{66}
$$

Hence, $\boldsymbol{m}^{(k)}$ is bounded.

**Proof (c):** See the proof of Theorem 1 (c).

For the case $r \in (0, 1)$, all conditions are valid. Hence, for the sequences $\{\boldsymbol{b}^{(k)}, \boldsymbol{u}^{(k)}, \boldsymbol{m}^{(k)}\}_{k=0}^{t}$ generated by the SCPRSM scheme (24a - 24d), and its Lagrangian given by (20), the three conditions from Theorem 1 (a)-(c) hold, achieving a worst-case convergence rate of $O(\frac{1}{k})$. Here, a worst-case $O(\frac{1}{k})$ convergence rate indicates that the solution's accuracy, based on specific criteria, improves gradually at a rate proportional to one divided by the number of iterations ($k$) within an iterative algorithm (He et al., 2014).

## C  Datasets

### C.1  Simulated QTLMAS 2010 Dataset

The dataset comprises 3226 individuals across 5 generations, including 20 founders (5 males and 15 females), with two observed traits (responses): a quantitative trait and a binary trait (Szydlowski & Paczyńska, 2011). Each female mates once, producing approximately 30 progeny per birth. SNP data were simulated using a coalescent model on five autosomal chromosomes, each 100 Mbp long. A total of 10031 markers were generated, including 263 monomorphic SNPs and 9768 biallelic SNPs. The continuous quantitative trait is controlled by 9 major QTLs at fixed positions, including two pairs of epistatic genes, 3 maternally imprinted genes, and two additive major genes with phenotypic effects of -3 and 3. The additive genes are positioned at SNP indices 4354 and 5327, whereas the major epistatic locus is at SNP 931. Additionally, a dominance locus was positioned at SNP number 9212, with an effect of 5.00 assigned to the heterozygote and 5.01 to the upper homozygote. Moreover, an over-dominance locus was placed at SNP 9404, with an effect of 5.00 assigned to the heterozygote, -0.01 to the lower homozygote, and 0.01 to the upper homozygote. After filtering SNPs with MAF $<$ 0.01, 9723 markers were retained and transformed into one-hot encoding, resulting in 29169 genomic markers. We used the quantitative trait in our study. Generations 1 to 4 (individuals 1 to 2326) were used for training, and generation 5 (individuals 2327 to 3226) served as test data.

## C.2 Real Pig Dataset

The Pig dataset contains data from 3534 individuals, with high-density genotypes and phenotypes for five traits (Cleveland et al., 2012). Using the PorcineSNP60 chip, 52842 SNPs were assessed and filtered to 50282 based on a minor allele frequency threshold of $< 0.01$. The chosen trait had a heritability of 0.58. After adjusting the phenotypic data and excluding individuals with missing data, the final dataset included 3152 individuals and was transformed into one-hot encoding, resulting in 150840 genomic markers.

## C.3 Real Mice Dataset

This dataset comes from an experiment aimed at identifying and locating quantitative trait loci (QTLs) associated with various complex traits in a population of mice. The dataset contains 1814 individuals who were genotyped for 10346 polymorphic markers and two traits: body length (BL) and body mass index (BMI). In this study, we used BL trait. After transforming the data into one-hot encoding, the dataset resulted in 31038 genomic markers. This dataset is from the Wellcome Trust and is available in the R package BGLR (Pérez & de Los Campos, 2014).

# D    Implementation of Baseline Methods

For comparison purposes, we implement the LASSO (3) using the proximal ADMM and SCPRSM schemes (referred to as LASSO-ADMM and LASSO-SCPRSM, respectively) as

$$
\begin{aligned}
\boldsymbol{b}^{(k+1)} &:= \mathrm{prox}_{f\gamma}(\boldsymbol{u}^{(k)} - \boldsymbol{m}^{(k)}), \\
\boldsymbol{u}^{(k+1)} &:= \mathrm{prox}_{g\gamma}(\boldsymbol{b}^{(k+1)} + \boldsymbol{m}^{(k)}), \\
\boldsymbol{m}^{(k+1)} &:= \boldsymbol{m}^{(k)} + \boldsymbol{b}^{(k+1)} - \boldsymbol{u}^{(k+1)}
\end{aligned}
\tag{67}
$$

and

$$
\begin{aligned}
\boldsymbol{b}^{(k+1)} &:= \mathrm{prox}_{f\gamma}(\boldsymbol{u}^{(k)} - \boldsymbol{m}^{(k)}), \\
\boldsymbol{m}^{(k+\frac{1}{2})} &:= \boldsymbol{m}^{(k)} + r(\boldsymbol{b}^{(k+1)} - \boldsymbol{u}^{(k)}), \\
\boldsymbol{u}^{(k+1)} &:= \mathrm{prox}_{g\gamma}(\boldsymbol{b}^{(k+1)} + \boldsymbol{m}^{(k+\frac{1}{2})}), \\
\boldsymbol{m}^{(k+1)} &:= \boldsymbol{m}^{(k+\frac{1}{2})} + r(\boldsymbol{b}^{(k+1)} - \boldsymbol{u}^{(k+1)}),
\end{aligned}
\tag{68}
$$

where $\mathrm{prox}_{g\gamma}(\boldsymbol{b}) = \mathcal{S}_\gamma(\boldsymbol{b})$.

Similarly, for EN-ADMM and EN-SCPRSM, $\mathrm{prox}_{g\gamma}(\boldsymbol{b}) = \frac{1}{1+\gamma\xi}\mathcal{S}\gamma(\boldsymbol{b})$, where $\xi > 0$ is a linear combination of the $L^1$ and $L^2$ penalties (Parikh & Boyd, 2013). The closed-form proximal mappings of the SCAD (5) and MCP (6) penalty functions can be found in (Fan & Li, 2001; Liao et al., 2023; Wang & Liu, 2024; Yun et al., 2021). Here, we utilize the scaled versions

$$
\mathrm{prox}_{g\gamma}(\boldsymbol{b}) = \mathrm{prox}_{\mathrm{scad}\gamma}(\boldsymbol{b}) = \begin{cases} \mathcal{S}_{\gamma\lambda}(\boldsymbol{b}) & \text{if } |\boldsymbol{b}| \leq (1+\gamma)\lambda, \\ \frac{(a-1)(\boldsymbol{b})-\mathrm{sign}(\boldsymbol{b})a\lambda\gamma}{a-1-\gamma} & \text{if } (1+\gamma)\lambda < |\boldsymbol{b}| \leq a\lambda, \\ \boldsymbol{b} & \text{if } |\boldsymbol{b}| > a\lambda, \end{cases}
\tag{69}
$$

$$
\mathrm{prox}_{g\gamma}(\boldsymbol{b}) = \mathrm{prox}_{\mathrm{mcp}\gamma}(\boldsymbol{b}) = \begin{cases} \frac{a\gamma}{a\gamma-1}\mathcal{S}_{\gamma\lambda}(\boldsymbol{b}) & \text{if } |\boldsymbol{b}| \leq a\gamma\lambda, \\ \boldsymbol{b} & \text{otherwise}, \end{cases}
\tag{70}
$$

with respect to SCAD and MCP, respectively. All iterations of LASSO-ADMM, LASSO-SCPRSM, EN-ADMM, EN-SCPRSM, SCAD-ADMM, SCAD-SCPRSM, MCP-ADMM and MCP-SCPRSM terminate upon achieving convergence, defined by the condition $\|\boldsymbol{b}^{(k)} - \boldsymbol{u}^{(k)}\|_\infty \leq \beta(1 + \|\boldsymbol{m}^{(k)}\|_\infty)$, where the tolerance parameter $\beta$ is set to $10^{-5}$.

