# OpenReview forum: "Weighted L1 and L0 Regularization Using Proximal Operator Splitting Methods"
_TMLR — Accepted by TMLR_

### Review · Reviewer_Vs8U · 2024-08-20

**Summary Of Contributions:**

This work proposes a joint weighted L1 and L0 norm regularization using proximal operators and translation mapping technique

**Audience:**

Yes

**Claims And Evidence:**

Yes

**Requested Changes:**

Please answer the questions above and include a new comparison

**Strengths And Weaknesses:**

Strength: the problem of L1 and L0 learning is important, and the datasets are nicely chosen

Weakness: There are quite a few points that I fail to understand, and the authors should have done a better job explaining them in the paper
1. I do not understand how the L0 part is being minimized. L0 is NP hard, and how does transforming equation (4) to Equation (12-13) helps minimizing the L0 penalty? This part is not explained. Also, since there is a theoretical complement, perhaps the theorem statements should state in a clear way how the optimization algorithm minimizes the L0 penalty

2. I am not sure about the comparison metric. Is the MSE computed on the test set? Or on the training set?

3. Why do the authors use ADMM to minimize LASSO? There are many optimization algorithms, why ADMM in particular?

4. I think think the authors should also compare with the method in https://arxiv.org/abs/2210.01212, where exact nonproximal algorithm for minimizing LASSO is proposed and may achieve much better performance here

5. The proposed algorithm seems so slow -- taking often more than a magnitude of time to run than any baseline. This appears to be a huge drawback to me

---

> ### Author Response · Authors · 2024-10-11
>
> Thank you for taking the time to review our manuscript and for your constructive comments. Below is our detailed response to each point (please note that the old equation numbers are followed by the new equation numbers in **bold**):
>
> 1. I do not understand how the L0 part is being minimized. L0 is NP hard, and how does transforming equation (4) to Equation (12-13) helps minimizing the L0 penalty? This part is not explained. Also, since there is a theoretical complement, perhaps the theorem statements should state in a clear way how the optimization algorithm minimizes the L0 penalty.
>
> Reply - The main idea is jointly minimize L1 and L0 using the translation function. Therefore, there is no alone L0 minimization. Transforming Equation (4) **(16)** to Equation (12) **(18)** is to form the constrained optimization problem and to solve it with the ADMM and SCPRSM frameworks. Equation (13) **(19)** is the objective function of the constrained optimization (Equation **(18)**, which is the Lagrangian function (Page 9, Section 4 Methodological Framework, and page 10, Section 5 Optimization Algorithms).
>
> 2. I am not sure about the comparison metric. Is the MSE computed on the test set? Or on the training set?
>
> Reply - The regression coefficients of the model are obtained from the training dataset, and once the model is trained, it predicts outcomes on the test dataset. The MSE is then calculated on the test dataset to assess the model's generalization performance (Page 13, Section 6.2 Results).
>
> 3. Why do the authors use ADMM to minimize LASSO? There are many optimization algorithms, why ADMM in particular?
>
> Reply - Since our focus is on applying a splitting method to the developed approach, ADMM is particularly well suited for this and therefore serves as the baseline method. Hence, for comparison purposes we also implemented Lasso in ADMM algorithm. It should also be noted that both ADMM and SCPRSM have the benefits of being very scalable and possible to parallelize which potentially could be useful for very large data sets. However, the focus in our paper has been on establishing the methodology, not on computational issues which we leave for future research.
>
> 4. I think the authors should also compare with the method in https://arxiv.org/abs/2210.01212, where exact nonproximal algorithm for minimizing LASSO is proposed and may achieve much better performance here.
>
> Reply - We have reviewed the referenced paper and included it in the related work section. While the approach in that paper is noteworthy, it aligns more closely with the study of sparsity in neural network frameworks. For comparison purposes, this may not fully align with our methodology. We appreciate your input and will consider this approach in our future research (Page 4, Section 2 Related Work).
>
>  5. The proposed algorithm seems so slow -- taking often more than a magnitude of time to run than any baseline. This appears to be a huge drawback to me.
>
> Reply - Thank you for raising this concern. We acknowledge that the running time of the proposed algorithm is an important factor. To address this, we changed to a constant learning rate instead of backtracking line-search, which has led to significant improvements in execution time. The running time of our method and the baseline methods has been improved.  Here, we report on the improvements in running time in our proposed method.
> From Table 1: The improved program for WL1L0-ADMM demonstrates a **2.89x speedup** over the original, reflecting a 65.44% reduction in execution time. Similarly, WL1L0-SCPRSM shows a **2.53x speedup** (60.47% reduction) (Page 14, Section 6.2 Results).
> From Table 2: The improved program for WL1L0-ADMM demonstrates a **3.75x speedup** over the original, reflecting a 73.35% reduction in execution time. Similarly, WL1L0-SCPRSM shows a **5.25x speedup** with an 81% reduction in execution time (Page 14, Section 6.2 Results).
> From Table 3: The improved program for WL1L0-ADMM demonstrates a **3.33x speedup** over the original, reflecting a 70% reduction in execution time. Similarly, WL1L0-SCPRSM shows a **2.04x speedup** with a 50.95% reduction in execution time (Page 15, Section 6.2 Results).

---

### Review · Reviewer_rSMH · 2024-08-25

**Summary Of Contributions:**

The paper presents a new model that combines both L1- and L0-norms (WL1L0) to enhance regularization in high-dimensional data. Despite the model's nonconvex and nonsmooth nature, the authors prove global convergence for several optimization methods, including ADMM, PRSM, and SCPRSM. The model's efficiency and the effectiveness of the proposed algorithms are demonstrated through experimental results.

**Audience:**

Yes

**Claims And Evidence:**

Yes

**Requested Changes:**

See weakness.

**Strengths And Weaknesses:**

Strengths:

1. This paper is well-written and easy to follow.

2. The new proposed model can take the benifit of both L1 and L0 norm and have good performance.

Weakness:

1. The problem (4) is derived from problem (2), and I am particularly interested in understanding the relationship between their solutions. Clarifying this connection would enhance the understanding of the model's development.

2. In the experiments, the parameter $\alpha$ is always small, implying that the L0 regularizer is more influential in ensuring sparsity. It would be valuable to compare the performance of your model against one that uses only the L0 regularizer.

3. Tables 1 and 2 show that ADMM solves problem (14) effectively and efficiently, with a shorter runtime compared to PRSM and a comparable minimum MSE. The inclusion of the parameter $r$ significantly improves performance when $r<1$. I recommend providing a more detailed explanation of the intuition and benefits behind SCPRSM.

4. I suggest introducing the K\L{} condition before presenting Theorem 1, as some readers might not be familiar with it. This would improve the clarity of the theoretical exposition.

---

> ### Author Response · Authors · 2024-10-11
>
> Thank you for taking the time to review our manuscript and for your constructive comments. Below is our detailed response to each point (please note that the old equation numbers are followed by the new equation numbers in **bold**):
>
> 1. The problem (4) is derived from problem (2), and I am particularly interested in understanding the relationship between their solutions. Clarifying this connection would enhance the understanding of the model's development.
>
> Reply - Thank you for your comment. We have tried to clarify and better explain this issue in the revised manuscript. See our revision of the problem in (4) **(16)** and (2)  **(14)** of the new manuscript (Page 7, Section 4 Methodological Framework).
>
> 2. In the experiments, the parameter α is always small, implying that the L0 regularizer is more influential in ensuring sparsity. It would be valuable to compare the performance of your model against one that uses only the L0 regularizer.
>
> Reply - In our WL1L0 implementation, the parameter α effectively controls the balance between the L1 and L0 regularizers. However, we have not conducted experiments using the L0 regularizer alone for direct comparisons because of convergence problems and its sensitivity to good starting values (Page 14-15, Section 6.2 Results).
>
> 3. Tables 1 and 2 show that ADMM solves problem (14) effectively and efficiently, with a shorter runtime compared to PRSM and a comparable minimum MSE. The inclusion of the parameter r significantly improves performance when r<1. I recommend providing a more detailed explanation of the intuition and benefits behind SCPRSM.
>
> Reply - In our revised manuscript, we have added a detailed explanation of the relaxation factor in SCPRSM and clarified its advantages compared to both the PRSM and the ADMM (Page 9, Section 5.2 SCPRSM Framework).
>
> 4. I suggest introducing the K\L{} condition before presenting Theorem 1, as some readers might not be familiar with it. This would improve the clarity of the theoretical exhibition.
>
> Reply - We have now introduced the K\L{} condition before Theorem 1 (Page 5 – 7, Section 3 Theoretical Background).

---

### Review · Reviewer_uxuY · 2024-09-14

**Summary Of Contributions:**

This paper proposes a novel regularization scheme that combines $\ell_0$ and $\ell_1$ regularization with the goal of retrieving relevant features (through the $\ell_0$ regularization), while at the same time preventing overfitting (through the shrinkage of $\ell_1$ regulization). The corresponding optimization problem is defined and then solved through three separate techniques (ADMM, PRSM, and SCPRSM), alongside convergence results. A disucssion about selecting the learning rate and tuning the hyperparameters is presented, and then the models are evaluated against baselines (consisting of LASSO, SCAD, and MCP using ADMM) on a synthetic and two real world genomics datasets.

**Audience:**

Yes

**Broader Impact Concerns:**

No broader impact concerns.

**Claims And Evidence:**

No

**Requested Changes:**

There are some changes that would greatly improve clarity. For example ADMM, PRSM, SCPRSM and SCAD and MCP are only introduced in the abstract. Similarly, SNP is only defined in the first few sentences of the introduction. These abbreviations are not used until much later, so when the discussion focuses around these terms it would help a lot in terms of clarity to restate them in full so the readers don’t have to go searching for them.

More on ADMM, PRSM, SCPRSM, why are you considering these optimization schemes? Howe are they different, what are the relative benefits of each, and why do you use all three? It seems there is no motivation to use them, and even a motivation for (12) (meaning, why was the problem infeasible before, how (12) makes it feasible) is missing.

I mentioned this above as well, but for someone with no significant biology background parsing and understanding even what the task is in the experimental section feels impossible. The descriptions are overly technical which doesn’t seem helpful for the main text and would be more appropriate in an appendix. It would help a lot readability if a simple description or explanation of the datasets and tasks was given and the technical details were moved to an appendix. As a further example of this, in the first experiment an MSE of $\sim 65$ is reported. In any experimental setting I’m familiar with, such an MSE would indicate a failure on the task. Making the experiments more understandable (and advocating why such a seemingly high MSE is actually acceptable) would go a long way.

Your conclusion states “The WL1L0 method outperforming all known regularization methods”. This is absurd as a statement, as you only test against three of them, not every known regularization method (as I mentioned before, strikingly comparisons against the elastic net are missing). I think it would strengthen your experiments a lot if you included comparisons against the elastic net and unregularized methods. Currently, the emphasis of the experiments is performance. If that is the main purpose, you should test against methods that could potentially outperform yours, otherwise the experiments should be designed in a way that is consistent with your motivation (sparsity, variable selection, overfitting).

**Comments**

Finally, I have some minor comments:

- Why is (29) defined? It’s identical to (27).
- Have you tried using accelerated versions of optimization, such as FISTA or Nesterov acceleration? These could potentially improve the running times of your methods.
- This might be a bit subjective, but 3.6 feels overly long to explain a hyperparameter choice.
- Appendices A, B, C are not named and are all named “Appendix”.
- What are the assumptions for the Figures 1, 2, and 3 in the appendix? It appears only the norms are plotted, not the error term, so that’s not $F$.

**Mistakes**

- The sentence “In cases where the variables are highly correlated, LASSO may randomly select variables” appears twice in the introduction. The sentence that follows the duplicate sentence is also extremely similar.
- Zhang (2010) above (9) does not have a softlink.
- Second paragraph in page 4, two closing quotations ($\textrm{'oracle property'}$ instead of $\textrm{`oracle property'}$).
- Page 9, in the first sentence of 3.6, “relazation”.
- Above the unnumbered equation before (57): “using the proximal operator for $g_2$ the prox of $g_2$ can be written as”

**Strengths And Weaknesses:**

**Strengths**

The paper is written in a straightforward manner and is easy to follow with minimal prerequisites from optimization. The structure makes intuitive sense and the ideas flow clearly.

In terms of content, the combination of $\ell_0$ and $\ell_1$ penalties is, to the best of my knowledge, novel. The theoretical results are straightforward and the inclusion of different optimization schemes doesn’t come off repetitive. Finally, the usage of the translation functions is interesting.

**Weaknesses**

There are some weaknesses in the work that can be mainly summarized in the following axes:

- unclear motivation,
- issues with the theoretical results,
- limited experiments.

*Unclear motivation*

It is not clear why we should be interested in using $\ell_0$ and $\ell_1$ together. In the introduction some motivation for this is introduced, but it is not convincing. For example, a reason the authors state for using sparsity is to reduce overfitting (first paragraph). In fact, in the second paragraph it is stated that $\ell_1$ reduces the magnitude of coefficients, and thus reduces overfitting. However, if the desire is to reduce overfitting, a (provably) better choice is $\ell_2$ regularization. Why is then $\ell_1$ chosen over $\ell_2$? Later in the introduction (top of page 2) it is stated that small coefficient values that are produced by LASSO are difficult to interpret and lack information. These statements seem to be conflicting, and moreover it’s not clear why the combination of $\ell_1$ and $\ell_0$ would avoid the stated problem of small coefficients. Finally, the elastic net is mentioned (but not compared against). It’s not discussed why the elastic net is not a good solution to this problem, which is rather strange, as it also has a very simple and efficient proximal operator.

*Issues with the theoretical results*

First, a comment on the $\ell_0$ component: in the second paragraph of page 2 a discussion is made about the NP-hardness of $\ell_0$. Then, at the top of page 3 it is stated that $\ell_0$ debiases the LASSO and because it follows LASSO then the computation is feasible. Unpacking that further brings up a few questions:

- What does, formally, debiasing mean in this context?
- Mentioning the NP-hardness of $\ell_0$ is peculiar to me. First of all, that NP-hardness is never addressed, in the sense of the authors showing how they overcome that hurdle. Second, mentioning NP-hardness seems irrelevant, since such results refer to global optimality. The authors instead use IHT which, if KL or RIP are satisfied, leads to stationary points, not global optima.
- Finally, it’s not the LASSO that makes $\ell_0$ “feasible”, it is IHT, and the feasibility itself could be questioned since the proposed methods are 4-20x slower.

In terms of Theorems 1 and 2, these seem standard proof schemes for Langrangians simply adjusted for the specific Langrangian of (13). There is a technical mistake in Theorems 1d and 2d: the statement says that global convergence is proven, but the actual proofs show convergence to a stationary point. Emphasizing more on this, it is unclear if (12)/(13) are general formulations for any $f,g$ or specifically for (4). The appear to be for (4), but in (19) the derivatives of the Langrangian are taken; at the very least the terminology is highly problematic as the $\ell_0$ does not admit a derivative, not even a subgradient.

Moving on to the implementation, the introduction of the translation functions is confusing. They are only discussed in A.4, but discussion reads that for affine transformations the proximal is a translation, and thus we can write a translation as $T(\boldsymbol{u}) = f(\boldsymbol{u} + \boldsymbol{m}) - \boldsymbol{m}$. It’s not obvious to me why this is true, why it is useful, or how it is used in (25). A proof that (24) is equal to (25) seems necessary here.

Finally, the mention of proximal operators in the context of $\ell_0$ seems misplaced. The authors frequently cite Parikh and Boyd (2013), but that and other monographs (Bach, Jenatton, Mairal, Obozinski, 2011) define proximal operators strictly for convex functions, which $\ell_0$ is not.

*Marginal experiments*

The experiments are hard to interpret without significant background in biology. Currently the description of the datasets is extremely technical and it is very challenging to even extract wha the tasks are. Regardless, the performance seems lackluster. In most cases the performance improvement is marginal and comes at a very high cost of 4-20x of increased computation.

Moreover, for LASSO, SCAD, and MCP ADDM is always used. However, in the authors’ methods it seems that SCPRSM is the method that performs more favorably. Given the small differences in performance, it is not convincing that the other methods wouldn’t perform better under similar optimization schemes.

Finally, the main motivation of the paper was to introduce $\ell_0$ in order to reduce the number of variables in the model, reduce overfitting, and improve variable selection. None of these aims are supported by the current experimental section, which focuses solely on MSE improvements.

---

> ### Author Response · Authors · 2024-10-11
>
> We sincerely appreciate you taking the time to review our manuscript and for providing such insightful and constructive feedback. Below is our detailed response to each point (the old equation numbers are followed by the new equation numbers in **bold**):
>
> Clarifying motivation:
>
> Reply - We are interested in combining L1 and L0 regularization to leverage the benefits of both regularization methods. L1 regularization is a convex minimization problem. However, L0 minimization is a difficult non-convex problem. The synergy of a combined L1 and L0 penalty offers a better balance between interpretability, predictive accuracy and computational tractability (Page 2, Section 1 Introduction).
> We have added a comparison of our proposed method against the Elastic net. This comparison reveals some similarities and some key differences. (Page 7, Section 4 Methodological Framework).
> We have also reorganized the sections and the text to get a more structured paper that hopefully has a better flow and is easier to read. In addition, we have added results for number of non-zero regression coefficients in order to provide some information about the sparsity and variable selection properties (Page 14-15, Table 1-3, Section 6 Numerical Experiments).
>
> Theoretical Results:
>
> Reply - Since L0 is an unbiased estimator and L1 often introduces bias in estimation, L0 can be considered as debiasing in our setting. This claim follows from the fact that all input variables are estimated with both L1 and L0 components from the b = (c+d) splitting and the translation mapping. Hence, the L0 part is added to the L1 part which counteracts the biasing shrinkage, but it doesn’t lead to fully unbiased estimates since there will always be an L1 part.
> We have removed the mention of NP-hardness from the text to avoid any confusion or irrelevant details. (Page 11-12, Section 5.5 Determining the Learning Rate).
> We have clarified in the manuscript that the proofs for Theorems 1 and 2 follow standard schemes for the Lagrangian in Equation (12)/ (13) **(18)/(19)** for Equation (4) **(16)**. Then Wl1L0 can be obtained by using translation function. Additionally, we have corrected the claims in Theorems 1d and 2d to indicate that the proofs establish convergence to stationary points (Page 7-10, Section 5 Optimization Algorithms).
> Regarding the sub-differentiability of L0, we have added Wu et al. (2021) for generalized subdifferentials of L0, with its regular subdifferentials provided in Le (2013) (Page 6, Section 3.1 Subdifferentials of Nonconvex and Nonsmooth Functions).
> We have clarified the section on translation functions in the manuscript, including Lemma 1 with a proof, which explains why it is useful and how it is applied. Additionally, we have corrected the references regarding the proximal operator of L0 (Page 7, Section 3.3 Proximal Operators, and Page 10-11, Section 5.4 Implementation).
>
> Clarification of Abbreviations:
>
> Reply - We have revised the manuscript to ensure that the abbreviations for ADMM, PRSM, SCPRSM, SCAD, MCP, and SNP have been defined upon first use in the text. We have also included instances of the full terms in later sections to enhance clarity for readers who may not be familiar with these concepts.
>
> Discussion of Optimization Schemes:
>
> Reply - We have added a section that outlines the differences between these methods, their relative benefits, and we have deleted PRSM and focused on ADMM and SCPRSM. ADMM is considered a baseline method that has the benefit from being easy to parallelize.
>
> Clarification in the Experimental Section:
>
> Reply - To improve readability, we have simplified the descriptions of the datasets in the main text, moving more technical details to the appendix. In the first experiment, the reported MSE of approximately 65 is due to the response variable not being scaled or normalized (Page 13, 6.1 Materials).
>
> Strengthening the Conclusions and Comparisons:
>
> Reply - We have revised the statement about the WL1L0 method's performance to more accurately reflect the scope of our comparisons. We have included additional comparisons against the elastic net (Page 16, Section 8 Conclusion).
>
> Use of Accelerated Optimization Methods:
>
> Reply - Our primary focus has been on applying a splitting method, with ADMM and SCPRSM being particularly well-suited for our approach. Therefore, we have not explored accelerated methods such as FISTA or Nesterov acceleration. We have improved the running times of the methods by using a constant learning rate instead of backtracking line search. (Page 14-15, Section 6.2 Results).
>
> We have named all appendices for better clarity and organization. Additionally, we have added detailed explanations of the figures to enhance understanding. We also removed Figures 2 and 4, as they did not provide additional value to the manuscript. All identified mistakes have been corrected in the manuscript (Page 20, Appendix A Related Problems).

---

### Decision · Action_Editor_nPBy · 2024-10-30

**Recommendation:** Accept with minor revision

**Comment:**

This paper proposes a new regularization approach that combines $l_0$ and $l_1$ regularization for preventing overfitting. The paper develops the theoretical convergence guarantee and provides the experimental results with comparison to various baselines. The authors have mostly addressed the reviewers' concerns about the mismatch between theoretical claims and the mathematical proofs, and have fixed errors in certain proof steps. Hence, I recommend to accept the paper.

However, the experimental results indicate that the proposed algorithms incur higher computational costs compared to the baseline methods. Thus, I request the authors to add a discussion about the computational limitations of the proposed method.

**Audience:**

Yes. The general topic of the paper will be interesting to a large body of TMLR's audience, whose interests are in optimization, sparse training, and high dimensional statistics.

**Claims And Evidence:**

The paper has developed both theory and experiments. The claims in the theorems are supported by the provided proofs, and the claims on the performance of the proposed algorithms are also supported by the experimental results.

---

> ### Author Response · Authors · 2024-11-18
> **Improved discussion**
>
> Thank you for the positive decision. We have expanded the discussion and added several important references for computational improvements that will be the focus of future studies.